

# IDENTIFICATION OF THE CONTRIBUTING AREA TO RIVER DISCHARGE DURING LOW-FLOW PERIOD

Gillet Maxime[1,2], Le Gal La Salle Corinne[1], Ayral Pierre Alain[2,3], Khaska Somar[1], Martin Philippe[3], and Verdoux Patrick[1]

[1]Université de Nîmes, unité de recherche Chrome France, F-30021 Nîmes Cedex 1
[2]Hydrosciences Montpellier, Univ. Montpellier, IMT Mines Ales, IRD, CNRS, Ales, France
[3]UMR 7 300 Espace CNRS, Université d'Avignon France

**Correspondence:** Gillet Maxime (maxime.gillet@unimes.fr)

**Abstract.** The increasing severity of hydrological droughts in the Mediterranean basin related to climate change raises the need to understand the processes sustaining low-flow. The purpose of this paper is to trial simple mixing model approaches first to identify and then quantify streamflow contribution during low-water periods. An approach based on the coupling of geochemical data with hydrological data allows quantifying flow contributions. In complement, monitoring during the low water period was used to investigate the drying up the trajectory of each geological reservoir individually. Data were collected during the summer of 2018 and 2019 on a Mediterranean river (Gardon de Sainte Croix). The identification of the end-members was performed after the identification of groundwater geochemical signature clustered according to the geological nature of the reservoir. Two complementary methods validate further the characterisation: rock leaching experiments and unsupervised classification (k-means). The use of G-EMMA mixing model coupled with hydrological monitoring of the main river discharge rate shows major disparities in the contribution of the geological units, showing a reservoir with a minor contribution in high flow becoming preponderant during the low-flow period. This finding revealed to be of the utmost importance for managing water resources during the dry period

## 1 Introduction

In relation to climate change, an increase in the severity of hydrological droughts, both in terms of duration and intensity, can be observed in the Mediterranean basin (Aubé, 2017). The increase in the severity of low water levels contributes to degrading the water resources both in terms of quantity, and quality (Nosrati, 2011; Chiogna et al., 2018) thus impacting ecosystems connected to the river (Folegot et al., 2018). This trend enhances to the need for a better understanding of the hydrological process during these periods of resource scarcity(Buytaert et al., 2006; Chiogna et al., 2018; Correa et al., 2017). Investigations on the process that sustains streamflow have been identified as a requirement to understanding the process that maintains low-flows (Smakhtin, 2001). Hence, as a first step, identifying the origin of the water that feeds streamflow during low water episodes is essential.

The approach often used in the study of low-flows targets the contribution dynamics of the different units of the watershed during those periods (Blumstock et al., 2015; Cartwright and Morgenstern, 2012; Cook et al., 2006) by focusing on the





differences of contributions amongst the major units of the watershed, i.e. shallow groundwater, deep groundwater, rainfall,

sub-surface, or on the exchanges with the water table in lowland areas (Petelet-Giraud et al., 2018; Blumstock et al., 2016). However, many studies emphasize the predominance of groundwater in maintaining flows in mountain areas (Tetzlaff and Soulsby, 2008) and more generally in maintaining base flow. It is also commonly accepted that the process of baseflow generation is controlled by the nature of the geology of the watershed (Bloomfield et al., 2009; Farvolden, 1963; Freeze and Cherry, 1979; Neff et al., 2005; Smakhtin, 2001; Tague and Grant, 2004). Some studies investigated the origin of water in a stream

based on the geological nature of the geological reservoir during high flow (Petelet-Giraud et al., 2018; Floriancic et al., 2018), but this has rarely been applied to low-flow due to the fact that many low-flow studies work on small basins with a strong geological homogeneity (Blumstock et al., 2015). The aim of the study was, therefore, to identify and then quantify the contributions of the different geological reservoirs during low water conditions in a watershed showing a variety of geological facies.

The method usually used to investigate the origin of water commonly conceptualise catchment areas in different landscape entities with specific geochemical signature and then unravel each reservoir contribution using hydrogeochemical mixing models, such as the End-Member Mixture Analysis (EMMA) (Christophersen and Hooper, 1992; Ali et al., 2010; Correa et al., 2017; Inamdar et al., 2013; Hooper, 2001). This approach considers the hydrogeochemical composition of the river water to be the result of the mixture of the different reservoirs contribution to flow (Christophersen et al., 1990). Assumptions

of conservative behaviour and linear mixing process are both equally necessary to run mixing models (Hooper, 2001). The contribution of each end-members is identified by tracing all potential water contribution to the stream flow, selected according to their ability to represent the overall variability of the geochemical signature of the stream data (Levia, 2011). The main interest of the EMMA analysis consists in the ability to consider the dispersion of tracers and thus consider all possible mixing configuration associate with model output probabilities (Barthold et al., 2017, 2011). With this tool, hydrogeochemical

information is particularly valuable when used in combination with hydrometric data (Buttle, 1994; Inamdar et al., 2013). It is made possible, for example, by differentiating water data based on season, independent rainfalls event to quantify the relative contribution of each member to the streamflow (Ali et al., 2010; Correa et al., 2017; Delsman et al., 2013; Inamdar and Mitchell, 2007; Inamdar et al., 2013; Morel et al., 2009). Model uncertainties are assessed based on the propagation of Gaussian errors (Genereux, 1998; Phillips and Gregg, 2001). Uncertainties in the contribution estimation obtained with these models can only

be minimised if the assumptions made for these tools are followed (Barthold et al., 2011). Estimates of the contribution of each end-member depend on tracers (Genereux, 1998), their numbers (Barthold et al., 2011), measurement uncertainties (Bazemore et al., 1994; Genereux, 1998) and the number of end-members included in the analysis (Delsman et al., 2013).

The majority of the studies using EMMA analyses focus on the identification of water during flood peaks (Brown et al., 1999; Burns et al., 2001; Engel et al., 2016; Evans and Davies, 1998; Fröhlich et al., 2008; Lloyd et al., 2016; Tetzlaff et al.,

2014; Tunaley et al., 2017; Yang et al., 2015). Some authors also worked on the whole hydrological year (Correa et al., 2019; Petelet-Giraud et al., 2018, 2016; Petelet-Giraud and Negrel, 2007) but the focus generally remains on floods rather than on low- flows. Most studies conceptualise water catchments into several major components: deep groundwaters, shallow groundwaters, soil water and rainwaters. By focusing only on low water in a watershed where the water table is limited,





it is possible to consider groundwater's contribution and differentiate the reservoirs according to their geology. This paper

search shows the applicability of these methods for identifying the origin of surface water during low-flow to understand flows dynamics in catchments during scarcity. This will provide a better understanding of the behaviour of this watershed during low-flow periods and allow to identify the reservoirs offering the highest productivity, which will ultimately allow improving management by focusing protection on these reservoirs.

This approach dealing exclusively with low water levels is of interest as, although the application of these methods is

frequent in hydrology, it is rarely applied during low-flow. The water contribution of each groundwater reservoir feeding the mainstream during the drying period will, first, be identified based on the geochemical properties of the reservoir and, second, be quantified. Then the drying-out curve of each of the reservoirs will be computed. Hence, in this present paper, we intend to identify the geological reservoirs contributing to river flows and then quantify their respective contributions during low-flow. Two strong assumptions are made: an exclusive origin of low-flow waters from groundwater and possible discrimination of the

end-members geochemical signature related to the geological formation. The proposed approach is applied to a real case study, and its effectiveness is demonstrated by combining different approaches in identifying the end-members and their signatures. Focusing on this period, and not on the whole hydrological year, allows a higher sampling rate and provides a finer analysis of the reservoirs' drying up mechanisms feeding the river (Floriancic et al., 2018). The article is organized into three sections. The first presents the watershed studied and the methodology proposed to identify the groundwater end-members in term of

the geological nature of the reservoir and then quantify their contributions. The second section describes the results obtained on identifying the end-members and on the contributions produced by the mixing models, while the third section provides a discussion on the followed methodology and results.

## 2  Methodology

This study aims at differentiating, during the low-flow period, the origin of the surface water according to the geological nature

of the contributing reservoir. This approach is based on the assumption that groundwater only supplies streamflow during low-flow periods. This allows us to exclude rainwater from the process and disregard these reservoirs, allowing us to restrain the number of contributing reservoirs, thus minimising the dispersion of our approach. The methodology relies first on the identification of different hydrogeochemical end-member present in the study area. These identified hydrogeochemical end-members are then linked to the different geological formations. Finally, a weekly hydrogeochemical survey of groundwater

and surface water during the summer allows us to quantify the contributions of each reservoir to surface water.

### 2.1  Study area

The study area is located in the south of France, in the *Cévennes* region, a Mediterranean mountainous chain 100 km North of Montpelier (see Figure 1 1). Our approach was developed on the Gardon de Sainte-Croix watershed, which presents an area of 95 km$^2$. This typical watershed of the Cévennes area presents relatively simple geology, with three dominant geological

units and limited anthropisation, which facilitate geochemical analyses and interpretation. The climate in the studied region





is defined as the Mediterranean with a very high annual rainfall of 1 110 mm average per year (Barre-des-Cevennes rain recorder, 1981 to 2010, Météo-France, noted BC on Figure 1). Though total rainfall is high, summer rainfall is very low, less than 50 mm, and almost half of the total annual rainfall falls in autumn during high-intensity rainfall events. The Gardon de Sainte-Croix river has a modulus of 0.960 $m^3$/s, and its Mean Monthly Annual Minimum discharge is equal to 0.135 $m^3$/s at

the hydrometric station located at approximately one-third of the basin length (*Pont Ravagers*, noted PR on Figure 11 ). The river is incised quite deeply into the relief showing fairly steep slopes. The altitudinal gradient is quite marked with an altitude ranging from 250 to 1 100 m over 30 km. From a geological viewpoint, the watershed is located at the beginning of the central zone of the Massif Central, showing a predominance of mica schists 1. A few granite stripes cross the upper section of the watershed, and a small limestone plateau forms the head of the basin. On the southern downstream slopes of the basin, mica

schists turn into black mica schists (Arnaud, 1999). Rocks are dated between Cambrian and Ordovician for basement rocks and between Bajocian and Hettangian for sedimentary rocks. In terms of land cover, the basin is composed of 90 % of forests and is sparsely populated with low agricultural activity showing less than 2 % of agricultural land. Inhabitants are about 1,200 on average during the year. Anthropic activities that can impact the stream water quality include tourism, with only two campsites and a cheese factory, all located in the downstream section of the basin. Hence the basin can be considered as little affected by

human activity and suitable to trial our approach carry out our research. Hydrogeological analyses in the watershed suggest the existence of a water body in the small limestone causse due to its slightly synclinal structure (Faure et al., 2009). The presence of this aquifer corroborates with the presence of a large number of springs at the limit of the sedimentary area. For the schistose part of the basin, no study suggests, to our knowledge, the presence of a water body in these areas. Only the small alluvial plains (very restricted in our basin) are likely to constitute an aquifer with limited capacity, directly connected to the live river

channels (Faure et al., 2009).

Low-flow values are severe in this watershed, with a discharge rate as low as 100 l/s, namely < 1 l/s/km$^2$, at the end of the dry season (Figure 2). Those low-flow levels occur rather late, with a minimum flow often found during September or October. The end of the dry season is determined by heavy autumn thunderstorms typical of the region—this study span over, two years 2018 and 2019. A large inter-annual variability can be detected in the period between 2017 and 2019. The year 2018 is notable

with a relatively high low-water flow, twice as high as in other years. Therefore, it can be assumed that the analysis of the contributions during these periods may shed some light on the differences in processes leading to this inter-annual variability in the flows. The volumetric discharge rate monitoring reveals that rainfall events have a very low impact on the river flow. It can be seen that discharge rate variations during the three years of observation are due to rainy episodes, but those present during the summer period lead to low peaks in a very short time. We note that during these events, the flow returns to a level

lower than that of the flow measured before the event in 1 to 3 days, which proves that the recharge brought by these rains to the subsurface reservoirs is negligible and makes it possible to disregard it in our future modelling.

## 2.2 Sampling and analysis

To identify the hydrogeochemical end-members, a prospecting campaign was carried out before the low-flow period between April and June 2018. Groundwater samples were collected at 17 sites in the watershed (see Figure 1). Boreholes were preferred,

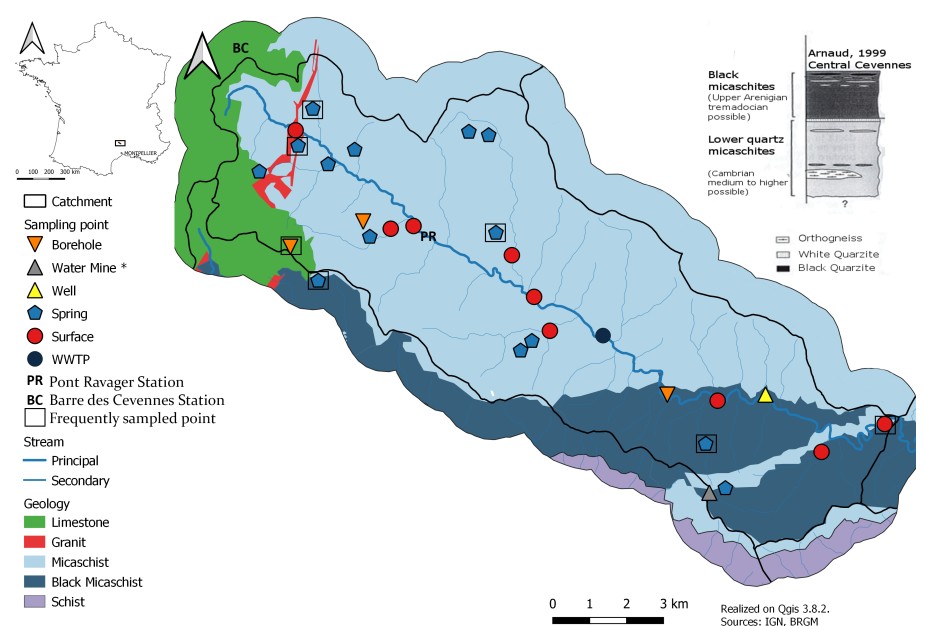

**Figure 1.** Geology of the Watershed studied: The Gardon of Sainte-Croix * A water mine is a horizontal well dug a slope.

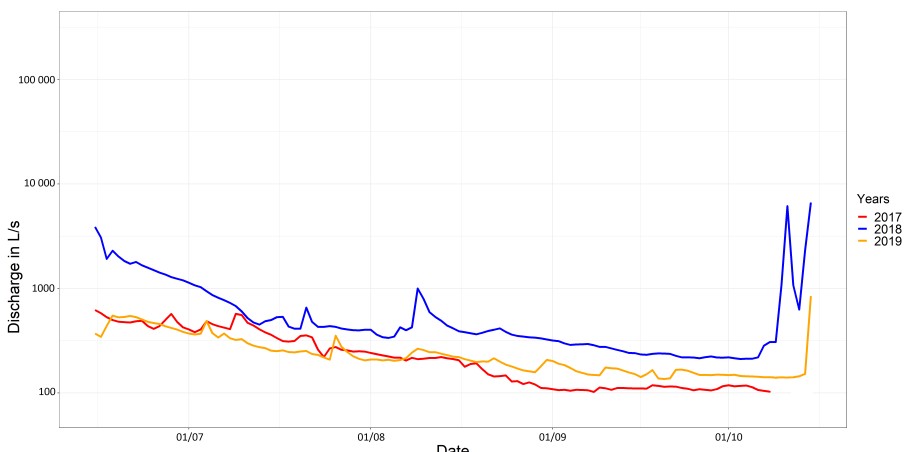

**Figure 2.** Gardon de Sainte-Croix hydrographs during the low-flow period between 2017to 2019. The end of low-flow period is different for each year and depends of the date of first important rainfall events. This hydrological data has been measured at the outlet of the catchment (Le Martinet Hydrometric station) by the UMR ESPACE 7300 CNRS (Martin et al., 2019).

but only a small number of relevant boreholes exist in the area, so most groundwater samples were collected on springs. The prospecting campaign was completed with existing data from the French National Groundwater Data Access Portal (ADES) to increase the number of observation points and consolidate the characterisation of the geochemical end-members. Physical





and chemical parameters (temperature, redox potential, hydrogen potential and alkalinity) were measured in situ at sampling sites. These measurements were carried out using a Hach SL 1000 multimeter. Temperature and electrical conductivity were measured with a CDC 401 probe, PH with a PHC 201 probe and redox potential with an MTC 101 probe. Alkalinity was also measured with a Hach multimeter using the reactive chemkey 8 636 200 for schist and granitic groundwater and 8 636 100 for limestone groundwater. Samples for the analysis of major ions were collected in 2 polyethene tubes (one for the cation and one for the anion). Water was filtered through a 0.25 $\mu$m cellulose acetate membrane filter. Tubes for the cation analysis were acidified to PH 2 with a drop of nitric acid titrated to 0.5 N and stored until analysis. The analysis was performed by ion chromatography (930 Compact ICFlex, Methrom). Major elements were carried out at the Laboratory of Environmental Isotope Geochemistry, University of Nîmes, EA 7352 CHROME.

For monitoring the low-flow period, the observation site was downsized to two representative sites for each reservoir. These points were selected based on the results of the prospective campaign and the identification of the geochemical end-members (Table 1).





**Table 1.** Sampling frequency table detailed over both summers. The bold row in the table correspond at main groundwater site sample weekly

| ID | Type | Outsourcing collection | Geology | Sampling in 2018 | Sampling in 2019 |
|---|---|---|---|---|---|
| **A2** | **Borehole** | **No** | **Limestone** | **9** | **11** |
| **A3** | **Spring** | **No** | **Mica schist** | **9** | **11** |
| **A4** | **Spring** | **No** | **Granite** | **10** | **11** |
| A6 | Water Mine | No | Black mica schist | 4 | 1 |
| C4 | Surface Water | No | | 16 | 16 |
| **E9** | **Spring** | **Yes** | **Black mica schist** | **11** | **7** |
| E10 | Borehole | Yes | mica schist | 11 | 0 |
| E11 | Well | Yes | Mica schist | 12 | 0 |
| F1 | Spring | Yes | Mica schist | 4 | 4 |
| F2 | Spring | Yes | Mica schist | 4 | 4 |
| F3 | Spring | Yes | Mica schist | 4 | 4 |
| G1 | Spring | No | Mica schist | 1 | 0 |
| G7 | Spring | No | Black mica schist | 1 | 0 |
| G8 | Spring | No | Limestone | 1 | 0 |
| **G12** | **Spring** | **No** | **Black mica schist** | **4** | **8** |
| G23 | Spring | No | Mica schist | 1 | 0 |
| H2 | Borehole | No | Mica schist | 1 | 0 |
| H4 | Spring | No | Mica schist | 4 | 0 |
| H5 | Spring | No | Mica schist | 1 | 0 |
| H6 | Spring | No | Mica schist | 1 | 0 |
| J1 | WWTP | No | | 0 | 1 |

The selection of groundwater site was made based on logistical reasons because not all sites could be monitored during the low-water period due to their non-perenity or poor accessibility. Spring with groundwater samples showing the influence of several geologies or boreholes located in the alluvial aquifer were also discarded from the monitoring to avoid bias in the characterisation of the end-members as they draw directly on surface water, and hence do not represent the geochemical signature of the local geological basement. Two monitoring campaigns were carried out during the summer of 2018 and 2019.

Both spanned from June to October; 6 springs and boreholes sites and one surface water point located at the basin outlet were sampled every week. The 2018 campaign focus on the characterisation of the groundwater contribution during the drying up period of the river with weekly monitoring, a greater frequency for surface water and bi-monthly for groundwater. The 2019 monitoring period was completed to include a spatial analysis where the stream was sampled on several sections (4), and





campaigns with a larger panel of groundwater were carried out every month, and the sample continued until December. The
frequency of sampling for this campaign was done every month, both for ground and surface water.

Water from the Wastewater Treatment Plant (WWTP) of the main village (Sainte-Croix-Vallée-Française, 350 inhabitants)
was also collected for analysing (Figure 1). In addition to monitoring, an additional campaign was carried out in 2019 to analyse
the spatial contribution of tributaries to the main watercourse throughout its route. Gauging and sampling were performed on
five sites distributed along the main river, and six tributaries were targeted (3 per side) using the same sampling and laboratory
analysis method presented above. The discharge measure was carried out by diluting gauging on the tributaries and exploring
the velocity field using a current meter for the main watercourse. The operation aimed to analyse the contribution of the
reservoirs with a spatial approach. However, only one tributary on the northern slope could be analysed as the two others were
dry.

## 2.3 Identification of end-members and selection of representative springs for low-flow surveys

### 2.3.1 Using groundwater analysis to characterise the end-members

The main assumption behind the geochemical approach is that the stream is a discrete mixture of the different groundwater
sources in the watershed. The samples analyses were categorised according to the reservoir geological nature and independent
statistical analysis based on different graphical representations. Two diagrams for graphical representation are used: the piper
diagram, which presents the relative concentration of major elements, and the bivariate solute-solute plots that show absolute
value results. end-members were defined by investigating the results from these two graphs, seeking to differentiate ground-
waters according to the geology of their reservoirs. To validate the identified hydrogeochemical end-members, a principal
component analysis (Christophersen, 1992; Long and Valder, 2011) is applied. A definition of end-members by classification
was also carried out. This was done by cluster analysis using "k-means", a classification method used in other studies (Fabbro-
cino et al., 2019; Monjerezi et al., 2011; Moya et al., 2015) to define end-members in a more complex system. Mean analyses
were based on major ion concentration normalised to the total dissolved solids to avoid dilution. The number of end-members
was defined by the average silhouette method defined by Rousseeuw (1987).

### 2.3.2 Validation of the end-member geochemical signature with a rock leaching experiments

To confirm the validity of defined hydrogeochemical end-members, a rock leaching approach was implemented. It aims to
strengthen the validity of the previously defined end-members by using an inverse approach. Rocks samples representative from
these formations are collected, and the rock leaching interaction experiment is carried out in the laboratory the geochemical
signature of the formation representing the geology (see Figure 3). This approach defines pristine groundwater and allows us
to eliminates end-member showing mixed signature between formations.

The leaching protocol was based on the widely used Afnor X31-210 standard and other articles (Chae et al., 2006; Gong
et al., 2011; Grathwohl and Susset, 2009; Yu et al., 2015). For this purpose, three rock samples were collected in each of the
identified geological units. Samples were all larger than 10 cm in size, a portion of each sample was set aside for rock sample





collection whilst the rest was crushed with tungsten beads and then sieved through a 4 mm mesh. Rock powder was mixed with ultrapure water (18.2 M) in a 50 mL bottle, in water-rock a ratio of 1/10 (3g rock water to 30g water). The leaching time was calculated through an experiment to obtain the rock water equilibrium. This test was performed only on a schist sample where a single sample was analysed at several different time steps (1 hour, 6 hours, 12 hours, 24 hours, one day, three days, one week,

two weeks, three weeks and four weeks) and the stabilisation of the major element point was obtained after three weeks. This bottle was then placed in a shaker for three weeks at 15 revolutions per minute. The result of this leachate was then analysed on the ionic chromatography machine presented in 2.2. Triplicates were made for each sample to improve repeatability and the accuracy of results. Four lithologies were sampled, limestones, granites, black mica schists and quartz mica schists. For each of these formations, three samples were taken from different catchment areas, except for granites, where their limited spatial

coverage did not allow multiple sampling sites.

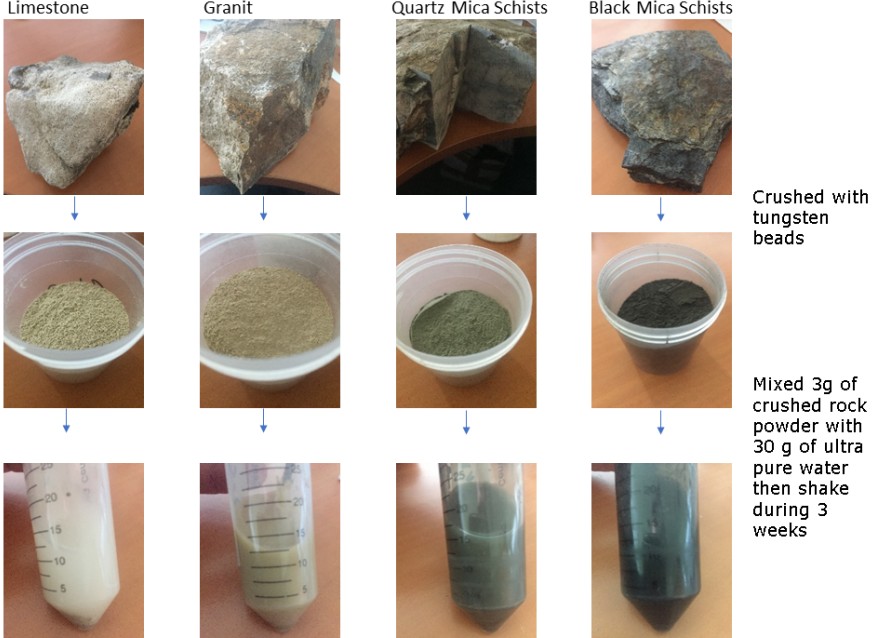

**Figure 3.** Rock leaching experimentation.

## 2.4   Mixing analysis

### 2.4.1   Choice of tracers for mixing analysis

Following the characterisation of the end-members, mixing models were implemented to estimate the contribution of each end-members to the streamflow. This model relies on a sound choice of tracers to calculate the part of mixing. Usually, two to six

tracers are considered depending on the number of considered end-members. They often include Ca, K, Mg, Na, Cl, $SO_4$ and





stable isotopes of water and physicochemical parameters such as electrical conductivity and alkalinity. (Barthold et al., 2011; Bresciani et al., 2018; Burns et al., 2001). In contrast to most research papers, usual conservative tracers are not considered in this study (the stable isotopes of water, bromides and chlorides), as conservative tracers are not affected by interactions with rocks, and hence cannot be used to differentiate the water according to their geological reservoirs (Appelo and Postma, 2005).

One of the study objectives is to test a simple method based on common and cheap tracers. Thus major elements are preferred to other more sophisticated tracers such as the strontium isotope ratio, for instance, used in other studies (Rose and Fullagar, 2005). The methodology to define the number of required tracers and the parameterisation for the use of the mixing model is based on a methodology developed by Barthold et al. (2011). This method first investigates correlations between the different tracers in order to eliminate redundant tracers and retain a number of tracers equal to the number of end-members plus one.

Tracers showing little variability or little correlation with the different end-members are also disregarded for this purpose.

### 2.4.2    end-member Mixing Model

The End-Member Mixing Analysis (EMMA) was chosen to assess the contribution of the different geochemical end-members identified. Our approach uses EMMA coupled with the Generalised Probability Uncertainty Estimate (GLUE), called G-EMMA (Delsman et al., 2013). This GLUE method, developed by (Beven and Binley, 1992), manages uncertainties by ac-
cepting variation in sets of input parameters. A full range of plausible results can be explored with model executions within a user-defined range by varying the input parameters. The G-EMMA method considers both the uncertainties in the conceptu-alisation of the model (validity of the choice of the end-members) and the measurement uncertainties related to the analytical errors. The variability accorded to the tracers chosen for the surface water is defined by the uncertainty associated with the devices used to analyse the data (5 %). A temporal variation treats uncertainties associated with the choice of geochemical
poles. Measurement uncertainties are defined by the variation in the geochemical signature of the end-members. Considering this variability makes the geochemical end-member approach more robust by providing results over the full range of plausible results.

In term of the model configuration, the number of iterations chosen was set at $10^8$. To solve mixtures, all defined end-members and all tracers must systematically be used. The option of "randomsolutes" was activated. It allows to vary randomly
the order in which the tracers are used in the modelling calculation randomly. To investigate the impact of the definition of geochemical end-members, four different methods were envisaged and compared. These methods are sorted in descending order according to their expected robustness and accuracy. The objective is to evaluate the loss incurred in the quality of results between these methods, which demand very distinct degrees of treatment:

- The first approach so-called hereafter "Time window (1)". Each end-member is defined by its concentration in elements
observed at a specific time in the groundwater and used to calculate the part of the mixture in the stream at the closest time measure of observation recorded in the watercourse (preferably before or if suitable just after the measure). The advantage of this method is to consider the seasonal variability of the solute concentration of groundwater.





– The second method, so-called hereafter "Seasonal Mean (2)", consider the mean seasonal value of the groundwaters selected as representative of the reservoir. Therefore, all mixtures are resolved using the annual average of the groundwater sites previously defined as representative of the formation. The variability given to these end-members is defined by the observed seasonal variability of the end-member.

– The third method, so-called hereafter "Geological Mean (3)", is based on an end-member signature defined by the average of the geochemical signature of all groundwater collected in the same geological formation for each reservoir without assessment of their representativeness. To give the same importance to each groundwater site, when some were sampled frequently whilst others were sampled just once, the average of the groundwater geochemical signature is calculated at each site before averaging the full results. The variability defines the variability given to these end-members observed in each of the formations.

– The last method, the so-called "Leaching Method (4)", uses the results of the leachate experiment and considers these results as representative of different end-members. End-members are simply calculated by averaging the three leachates carried out for each formation. Due to the relatively small number of samples, the variability of these end-members is defined by the variability of the results added to the ion chromatography analysis results (5 %).

## 3 Results

The results of the end-members' characterisation and validation are presented first, then the results of the mixture models are exposed in the second section. The characterisation of the end-members is based on the analyses of the collected groundwater samples. The K-means classification and leaching approaches presented above are then considered to consolidate the end-member clustering. Results of the mixing models are subdivided into four steps:

– The selection of appropriate tracers

– The presentation of model results based on the first method, named the "time window"

– The comparison between the results of different model output (time window, seasonal mean, geological mean and leaching method) based on varying approaches for the characterisation of the end-members

– Analysis at the catchment scale of the contributions

### 3.1 Identification of the end-members

#### 3.1.1 Identification of the end-members by groundwater analysis

On the piper diagram, three end-members are identified (see Figure 4): The first one is marked by a magnesium and calcium signature for cations and bicarbonate for anions. This end-member is composed exclusively of water from sedimentary rock reservoirs, mainly limestones and dolomites, hence consistent with the composition of 3 groundwater found in the literature



(Clark and Fritz, 1997). This end-member is also identifiable on the bivariate solute-solute diagram where we can see that these waters have conductivity values much higher than other end-members, ranging between 400 and 450 $\mu s$/cm, while most of the others are below 100 $\mu s$/cm (see Figure 5). This high conductivity is related to high concentrations of three elements, calcium,

magnesium and bicarbonate (3, 2, 5 MEQ/l), while concentrations of other elements remain relatively low.

A predominance of sulphate for anions marks the second end-member. The signature for cations is relatively undifferentiated but tends towards a slightly more magnesian facies. This leads to groundwater marked by a predominance of sulfates ions located in water hosted in the black mica schist formation. However, all springs sampled in this formation do not systematically show an excess in sulphate. Indeed, sulphate contents vary from 0.3 to 1 MEQ/l and remain relatively low for all other elements.

According to previous studies, this sulphated signature could result from schists alteration (Mayer et al., 2010).

Groundwaters from the third pole come from quartz mica schists reservoir. The end-member shows a large dispersion with an undifferentiated signature. These waters are characterised by a very low conductivity (less than 60 $\mu S$/cm) and very low concentration in all elements, which strongly differentiates them from other analysed end-members (Figure 5). Therefore, this end-member can be considered as undifferentiated, i.e. no element is present in greater proportion than the others. The observed

dispersion of the signature on the piper diagram can be explained by the very low concentrations of elements, leading to a large variation of the geochemical facies due to only small variations of individual element concentrations.

Also presented in Figure 4 and 5, a unique groundwater sample collected in the granite show surprisingly high bicarbonate, calcium and magnesium content (2, 1 and 1 MEQ/l) and also, to a lesser extent, the presence of sulphates. These concentrations place this sample on the mixing line between two previously defined end-members, the limestone and the black mica schist end-

members. The influence of the limestone end-members seems coherent because of the topography and stratigraphic position of the granitic layer crossing the limestone plateau. Moreover, drillings in the area show that the black mica schist layer is present just below the limestone plateau. It is, therefore, possible that springs collected in the granite sections are, in fact, water that percolated through the limestones and then the black mica schists.

A seasonal evolution in ion concentrations is clear in both the waters of the black mica schists and the limestones (see Figure

5). During the drought period, ions concentrations increase, particularly visible for Ca, Mg, $HCO_3$ and $SO_4$. This increase in concentration is 12 % for the limestone waters and 20 % for the black mica schists waters.

Finally sodium facies marks the water from wastewater treatment plants for cations and high concentrations of sulphates and chlorides for anions with a relatively high electric conductivity (350 $\mu S$/cm).

### 3.1.2   Rock leaching experiment

The results of the rock leaching experiments lead to leachate geochemical signatures quite close to theses observed for groundwater samples analyses. The differences observed between groundwater and leachate water are of the order of 20 % in conductivity with lower concentrations on average for leachate water. Four end-members of lixiviation are visible on the bivariate solute-solute diagrams of groundwater and leachate samples (Figure 6). These results can be summarised as follow:

     – Black mica schist leachates show a high proportion of sulphates and higher magnesium than calcium contents.



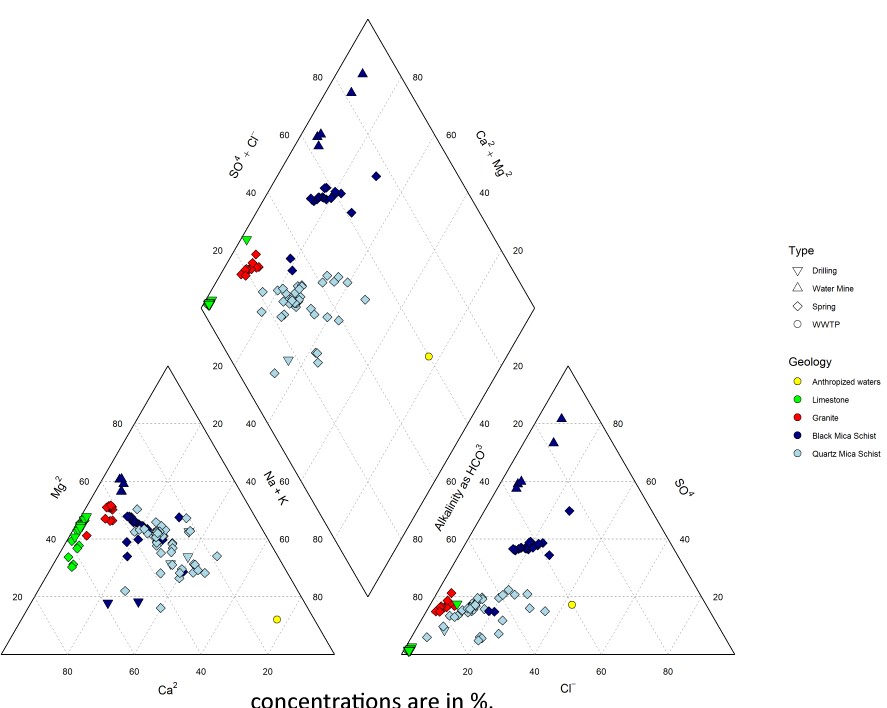

concentrations are in %.

**Figure 4.** Piper diagram of groundwater sampled.

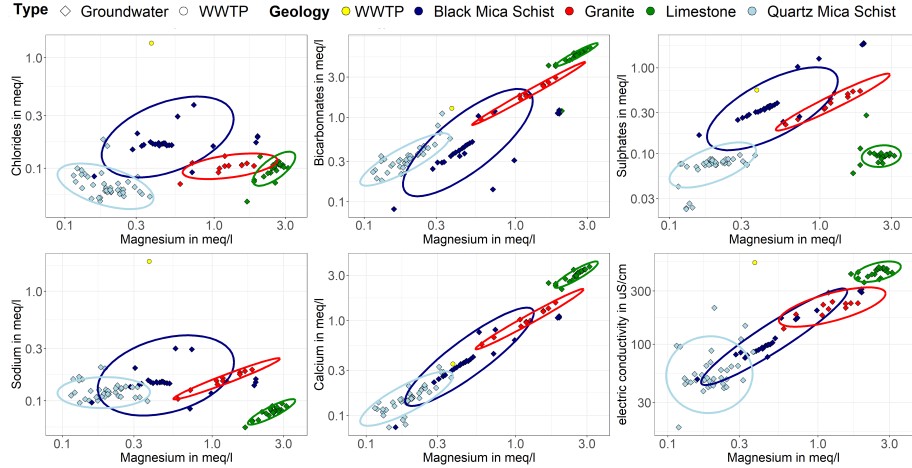

**Figure 5.** bivariate solute-solute diagrams of groundwater. Ellipse in the graphs was calculated with stat ellipse (ggplot2 package).

–   Quartz mica schists leachates are defined by a neutral signature with conductivity lower than 30 $\mu$S/cm.





– Limestone leachates are marked by higher Ca, Mg and $SO_4$ content. However, values observed in leachates (1 MEQ/l) are three times lower than those observed in springs and boreholes (3 MEQ/l). This may be inherent to the leaching experiment carried in a closed bottle with a limited quantity of $CO_2$. Indeed the lack of exchange with the atmosphere during the leaching process could limit the concentration of dissolved elements in the leachate (Appelo and Postma, 2005).


– A granite pole marked by the presence of sodium and potassium in very large quantities can be identified here. The obtained leachate geochemical signature differs to that observed for the groundwater collected in this formation (showing both low Na and K content).

Larges amount of potassium is observed in the leachates of the crystalline rock samples typical of the weathering of potassium feldspars (Appelo and Postma, 2005; Clark and Fritz, 1997). Since no collected water shows this signature marked in sodium and considering that this layer has a very small spatial footprint, this granitic reservoir is disregarded in estimating the contributions to streamflow.


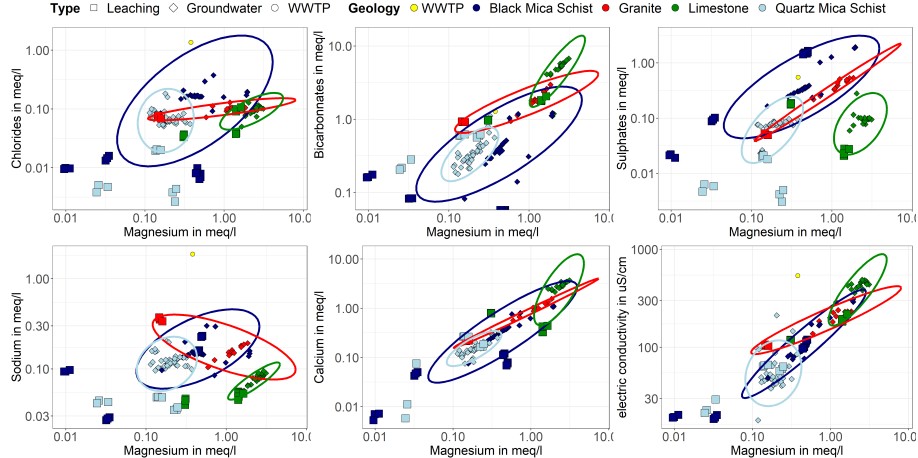

**Figure 6.** bivariate solute-solute diagrams of leachate result with groundwater sample .

Leachate results raise questions regarding the relatively large amount of potassium in the metamorphic rock samples. These quantities are three times higher than those observed in the groundwaters. These differences can be explained by the leaching method (crushing of the rocks to a very fine granulometry), which favours the potassium solution via the alteration of potassium feldspars (Appelo and Postma, 2005; Clark and Fritz, 1997) while K in situe may already have been lixiviated.


### 3.1.3 Validation of the end-member by statistical classification

In order to confirm the end-members' characterisation and clustering independence, a statistical approach has been carried out on collected groundwaters. Focussing on the groundwater end-member analyses, the WWTP's water was not included in the



statistical analysis. The first step in this method is to define the number of clusters. The inertia curvature of groundwaters shows
in both cases that the optimum number of classes is three (Figure 7). This number of three classes corresponds to the number
of identifiable end-members found in the catchment. This number is to be used for upcoming k-means analyses.

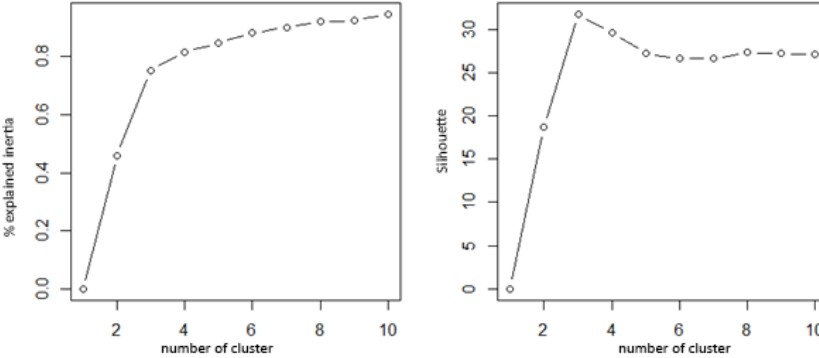

**Figure 7.** Silhouette Values to define the optimum of cluster .

Inertia curves define the k-means method based on the 3 classes, gives equivalent results to previous analyses on groundwater
samples to characterise the end-members. Moreover, the three clusters defined by this method correspond to the three identified
end-members and hence to the three main geological reservoirs, namely limestone, quartz mica schists and black mica schists.

Indeed, the first cluster is defined by a low conductivity and high proportion of bicarbonates in the water, which is coherent
with the quartz mica schists reservoir (Table 2).

The second cluster is defined by high sulphates and magnesium proportion and corresponds to the black mica schists reser-
voir. The third shows a high proportion of calcium, magnesium and bicarbonates in the water, and high electrical conductivity,
which is coherent with the limestone reservoir.

The location of the groundwater samples, identified by clusters, are plotted on the geological map showing the good corre-
spondence between the 3 clusters and the three geological reservoirs (see Figure 8). Only one outlier is visible and corresponds
to the point presents in the granite formation and previously identified as a mixture of limestone and black mica schist. Con-
versely, the K-means method attributes this point to the sedimentary rock clusters' in coherence with the mixing hypothesis of
groundwater issue from limestone and black mica schists.

**Table 2.** Mean proportion in % of the major elements in the cluster result

|  | **Ca** | **Mg** | **Na** | **Cl** | **SO$_4$** | **HCO$_3$** | **EC** |
| --- | --- | --- | --- | --- | --- | --- | --- |
| **Cluster 1** | 32 | 33 | 28 | 22 | 15 | 63 | 15 |
| **Cluster 2** | 32 | 47 | 19 | 19 | 53 | 28 | 25 |
| **Cluster 3** | 54 | 42 | 4 | 4 | 5 | 90 | 82 |



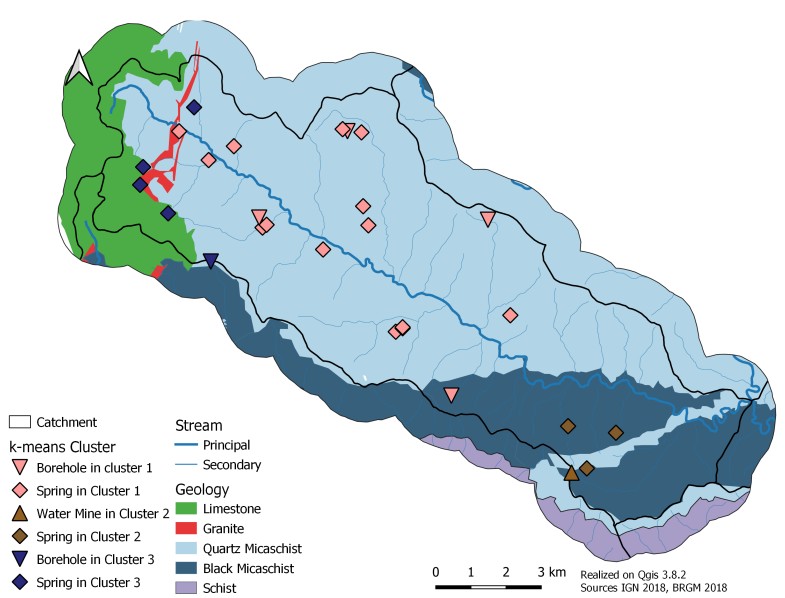

**Figure 8.** Clusters localisation obtained with k-means method.

## 3.2 Mixing results

### 3.2.1 Choice of tracers

Regarding the choice of tracers used in the mixing model, previous studies as (Barthold et al., 2011; Christophersen, 1992)
recommend disregarding those with too strong inter-correlation or too weak variances. Principal component analysis (Figure 9)
shows the strong correlation between the limestone end-members (in green) with the $Ca^{2+}$, $Mg^{2+}$, $HCO_3-$ and $Sr^{2+}$ tracers.
Black mica schist (in navy blue) is significantly connected to $SO_4^{2-}$ and to a lesser extent to $Cl^-$ and $F^-$. The pole of quartz
mica schist (in cyan) shows a very low variance with both axes. However, it shows a slight inverse correlation with the axis 2
which variance is explained by $SO_4^{2-}$.

    Based on those observations selected tracers are $HCO_3^{2-}$, $SO_4^{2-}$, $Mg^{2+}$ and $Na^+$. $HCO_3^{2-}$ was selected for its correlation
with the limestone end-members and $SO_4^{2-}$ for its correlation with the black mica schist end-member. Sodium was also se-
lected due to its connection to the wastewater pole. Minor ions have been disregarded due to their low frequency of detecting
concentrated water, particularly for fluorides on $Ca-HCO_3$ water. Due to its low concentration in the totality of the measured
elements, no tracers are specifically designed for quartz mica schists since it acts as a dilution pole for all tracers. Low miner-
alisation in all tracers is the marker of this end-member. To improve the efficiency of the model in the contribution calculation,
the choice was made to add tracer to the tracers chosen by the end-members. Because of their strong explanation of variance,
calcium and magnesium were selected, but with the high correlation between these elements (Table 3), only magnesium is





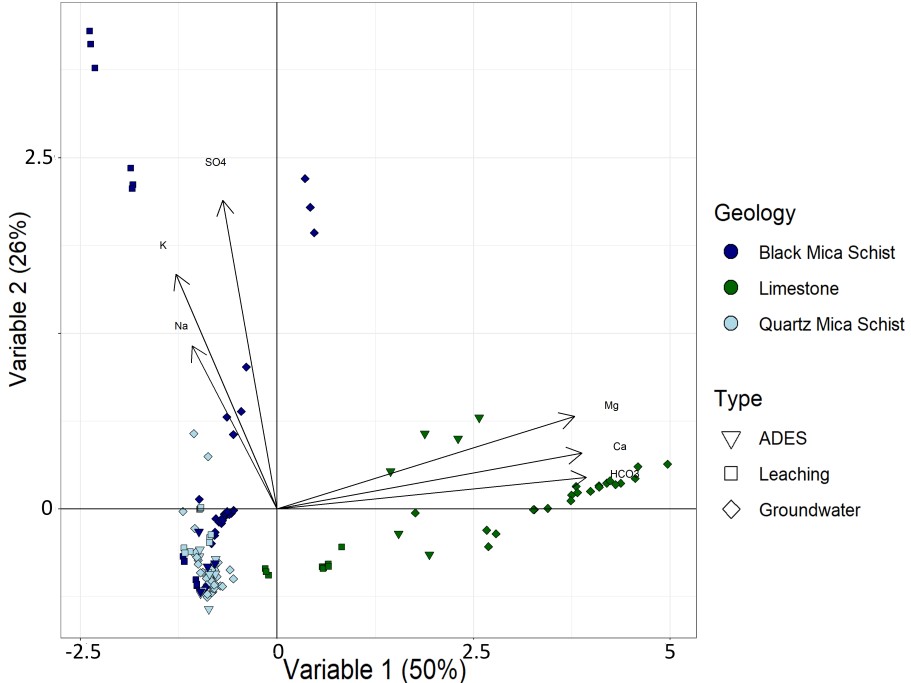

**Figure 9.** Principal component analysis .

selected to limit the weight of the calcareous water contribution to the mixtures. Magnesium is preferred at calcium because it is slightly correlated to the pole of black mica schists making it more relevant and different to Bicarbonates (Figure 9). These selected tracers have the particularity of being reactive in groundwater reservoirs, allowing them to be marked by the passage
in this reservoir but can be considered as conservative in the watercourse. Indeed, in the stream, water-rock interactions are reduced, and equilibrium is rapidly obtained with the atmosphere. The measurement of dissolved oxygen in the springs and surface water confirms this by revealing identical oxygen concentrations to those found in the streams and springs.





**Table 3.** Correlation matrix. The red values show the correlation greater than 0.5.

|        | PH    | EC    | Ca    | Mg    | Na    | K      | St    | Li    | Cl    | SO$_4$ |
|--------|-------|-------|-------|-------|-------|--------|-------|-------|-------|------|
| **PH** | 1     |       |       |       |       |        |       |       |       |      |
| **EC** | 0.31  | 1     |       |       |       |        |       |       |       |      |
| **Ca** | 0.36  | **0.96** | 1  |       |       |        |       |       |       |      |
| **Mg** | 0.33  | **0.96** | 0.97 | 1   |       |        |       |       |       |      |
| **Na** | -0.44 | -0.29 | 0.33  | -0.30 | 1    |        |       |       |       |      |
| **K**  | -0.11 | 0.02  | 0.33  | -0.03 | 0.20 | 1      |       |       |       |      |
| **St** | 0.33  | 0.36  | 0.34  | 0.41  | 0.08 | 0.03   | 1     |       |       |      |
| **Li** | -0.33 | -0.09 | -0.16 | -0.7  | 0.01 | 0.01   | -0.03 | 1     |       |      |
| **Cl** | -0.49 | 0.04  | -0.01 | 0.01  | **0.83** | 0.18 | -0.17 | 0.14 | 1     |      |
| **SO$_4$** | -0.32 | 0.11 | -0.02 | 0.18 | 0.18 | 0.09 | 0.08  | **0.56** | 0.30 | 1 |
| **HCO$_3$** | 0.41 | **0.93** | **0.98** | **0.94** | 0.34 | -0.003 | 0.38 | -0.22 | -0.07 | 0.10 |

In this analysis, it is evident that it is impossible to differentiate waters of quartz mica schists and rainwater. Indeed, rainwater
collected in the area with an electrical conductivity of 14 µS/cm is only slightly lower than that of the mica-schist water, has
an average electrical conductivity of 44 µS/cm². Moreover, rainwater shows an undifferentiated signature, similar to the water
from the quartz mica schists reservoir. Hence, this model must be used exclusively in low-flow conditions so that the proportions
of water identified as issued from quartz mica schists are not confused with the portion issued from rainwater.

### 3.2.2   Result of mixing analysis

The results obtained using the G-EMMA with the "time window" method are shown in Figure 10. To start with, it is noticeable
that both summer periods, 2018 and 2019, differs strongly in terms of hydrology. The Gardon de Sainte-Croix discharge varied
from 600 to 200 l/s in 2018 and 300 to 150 l/s in 2019. These differences in flow rates can be explained by higher cumulative
precipitation in spring 2018 (700 mm) than spring 2019 (375 mm). This difference in the amount of precipitations is interesting
as it allows for comparisons of the behaviour of this river system both during low-flow and slightly more severe drought period.
Nevertheless, the mixing model gives overall relatively similar results for both summers. A relatively limited contribution from
sedimentary rock reservoirs(-10 %) is observed, while the largest part is coming from schist rocks (90 %). The quartz mica
schists and black mica schists contribute roughly in the same proportion at the beginning of the summer(see Figure 10), then
a decrease in the contribution of quartz mica schists and a relative increase of the contribution of black mica schists is evident
toward the end of the dry season. WWTP effluents show an extremely low contribution between 1 to 2 per cent. A more
important contribution of WWTP can be observed from mid-July to the end of August, coherent with the increasing tourism
activity within the watershed but remaining below 4 %.

The distribution of the calculated contributions allowed by the model is very low for the results on water contributions from
the limestone end-member and WWTPs, remaining below 1 % of the contribution. They are greater for schist waters, ranging





from 10 to 25 %. The relative similarity in mineralisation between the two end-members (Quartz mica schist and black mica

schists) and their dispersion leads to a wider range of possible results.

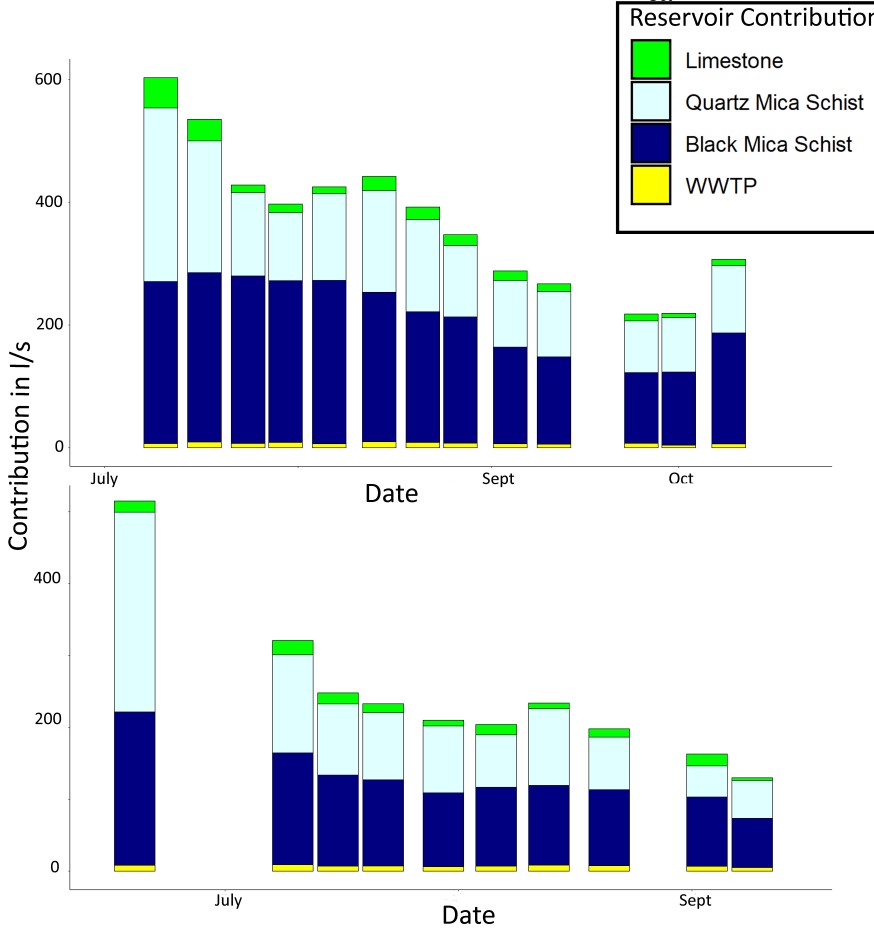

**Figure 10.** Evolution of the contribution of the various reservoirs during the summer of 2018 and 2019. The difference in the number of samples between 2018 and 2019 is due to the differences in sampling frequency between the two years, weekly and half monthly for 2018 and for 2019 respectively.

At the beginning of the monitoring period in 2018, the quartz mica schists bring 290 l/s and drop to 75 l/s in low-flow, while the black mica schists reservoir contribution varies only from 270 to 130 l/s. In 2019, the quartz mica schists contribution brought 120 l/s initially and dropped to 30 l/s at the end of the low-flow period whilst black mica schists flow only drops from 160 Ls to 90l/s.

Expressed in % of the flow rate, in 2018, the contribution to the flow rate is about 45 % for the black mica schists reservoir and 50 % for quartz mica schists. At the end of the summer, the contribution of quartz mica-schists reservoir drops slightly to 30 % while the black mica schists reservoir provides 65 % of the flow. For the year 2019, the contribution is already unequal





at the beginning of the season between the two formations, with nearly 55 % ensured by the black mica schists and 45 % for the quartz mica schists. The relative contribution of the black mica schists reservoir increases significatively during low-
flow conditions, where it reaches 75 % of the total flow. The limestone reservoir shows a relatively low contribution whilst remaining relatively constant with a value between 5 and 10 % throughout both summers. Hence, at the beginning of the summer (June 2019), most of the water flow comes from quartz mica schists, while the contribution of the black mica schists become preponderant in low-flow. Surprisingly, the contribution of the black mica schist reservoir is very high for the small surface area of this formation outcrop, approximately 20 % of the surface area. The decreasing flow rate is very different
between both schist reservoirs. They show relatively equivalent flows at the beginning of the season, which decreases during the dry season, with a reduction of the flow by 4 for the quartz mica schists and only by 2 for the black mica schists.

The drying curve of these two reservoirs is consequently very different, reflecting two very different behaviours with a much steeper slope and therefore demonstrating a much lower low water production capacity for the quartz mica schists during the low-flow period. The specific flow rates calculated with respect to the outcropping surfaces of each geology are less than 1
l/s/km$^2$ for quartz mica schists and more than 2 l/s/km$^2$ for black mica schists. All this underlines the importance of the black mica schists reservoir supporting low-flow levels worsening in the extremely low-flows period.

### 3.2.3   Uncertainly of mixing analysis

To compare the results obtained with the different approaches considered for end-members signature characterisation, the outputs of the four models (Time window, seasonal mean, geological mean and leaching experiment) were plotted together in
Figure 11. All four approaches give similar results and trends than that observed with the "Time window" method. Dispersion in the contributions remains high, reaching a variation of 25 % between the two quantiles and nearly 50 % between the limit of the models: time window and seasonal mean present low dispersion in the range of possible contributions. However, general trends seen on contribution graphs remain identifiable and consistent from one model to another, with an increase in the contribution of black mica schists and a decrease in the contribution of quartz mica schists during summers. The first autumn rains can
explain the steep increase in the contribution of quartz mica schists and the decrease in black mica schists' contribution at the end of the season. The autumn rains reverses the contribution of these two reservoirs as they recharge the quartz mica schists' reservoirs, which are much larger in surface area than those of the black mica schists and directly diluting the river water, acting, as a contributing of the quartz mica schists reservoir diluting the surface water.

Differences are nevertheless visible between the four outputs: The selection graph by geology, which uses average values of
all water collected in the same formation, shows the greater variability for the contribution of the quartz mica schists reservoir. This variability can be explained by a larger dispersion in waters signature encompassing all groundwater analyses over the observed period, thus integrating seasonal variations and leading to the definition of less constrained end-members. This gives the model greater freedom to solve mixtures.

The "leaching method" also shows less constrained outputs. These are mainly visible on the contribution of the limestone
pole, of which the contribution is more important than for the other three models outputs. This relates to the fact that limestone leachate end-member is artificially less concentrated than for other approaches, leading to an overestimation of its contribution.





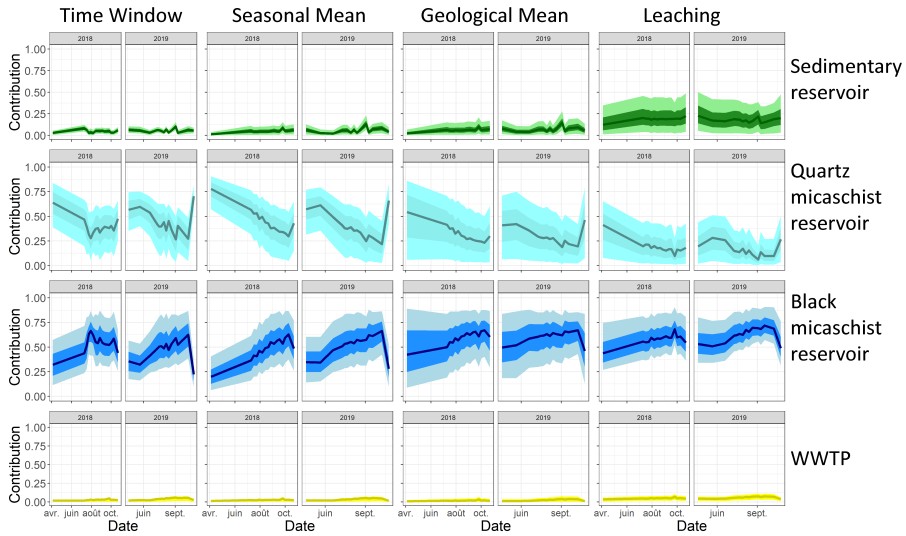

**Figure 11.** Mixing models result uncertainties for different contributions. The green colour represents the contribution of the limestones, the cyan represents the contribution of the quartz mica schists, the dark blue represents the contribution of the black mica schists and the black represents the contribution of the WWTP waters. For each variable the central line is the median value of the model, the outer limits are the limits of the model (respectively 5 and 95 %) and the limit of the darker colours corresponds to the quantile of the model. The contributions are expressed in index ranging between 0 and 1, where 1 corresponds to the sum of the contribution.

There is also a difference between the "Time window" method and the "Seasonal mean" method while the signal appears smoothed. This difference can be explained by the account being taken or not of the seasonal drift (see Figure 11). Regarding the "seasonal average" output, the results show a lower contribution of the waters with the highest low-flow electrical conductivity and a higher contribution of the waters with the lowest loading.

### 3.3 Spatial analysis to modelling results

The investigation of the spatial distribution of the different reservoir contributions was carried out. This spatial approach consisted in collecting samples and measuring the flow rate along the river length along with the main tributaries on the same day. This campaign was carried out during the 2019 low-water period (October 10th). At that time, the measured flow rate was 142 l/s, while this year's lowest flow rate was 135 l/s. The measurement was performed on the three biggest tributaries in the right and left banks. Only one of the targeted tributary was surveyed on the left bank, as all other streams had dried out.

The results underline the black mica schists' predominant contribution to low-flow throughout the watershed (see Figure 12). The results also show an uneven spatial distribution of the specific flow rate. The mainstream specific discharge varies widely from the headwater to the outlet, with more than 2 l/s/km$^2$ at the most upstream section (station 1, 6 or 8) decreasing to approximately 1 l/s/km$^2$ at the outlet (station 5). Regarding the tributaries, the differences are even greater with specific flows of less than 0.1 L/km$^2$ on the Northern slope (left bank, station 7),and reaching nearly 2 l/s/km$^2$ on the Southern slope (right





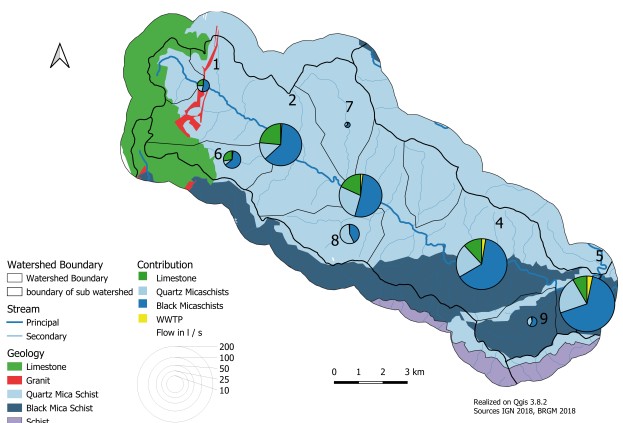

**Figure 12.** Map of contribution of the different aquifers in the Gardon de Sainte Croix basin on 11th October 2019.

bank,6,7,9). The contribution of the upper limestone reservoir remains nevertheless a minor contribution (< 20 %, at station 1 or 6) and cannot explain the observed upstream high flow rates. It is noticeable that the upstream flow already relies heavily on the black mica-schists and quartz mica-schists reservoirs. The high upstream and southern slope specific flow rates may

be explained by the presence of a black mica schist stratum, identified as the main source of water during low water levels, located below the upper limestone plateau and extending on the southern slope (Arnaud et al., 2004). .In terms of contribution levels, the black mica-schists reservoir remains the main contributor throughout the basin. The quartz mica schists reservoir contributes only as a secondary source on the order of 25 % of the low-flow rate, except on a tributary of the south slope located downstream of the watershed. The wastewater treatment plant (WWTP) contribution is insignificant on small tributaries and

increases slightly down steam along the mainstream with increasing urbanisation.

## 4 Discussion

### 4.1 Are the identification of mixing poles and the significance of geochemical poles correct?

Mixture models are powerful tools that can deal with many scenarios and thus provide a wide range of possible solutions (Soulsby et al., 2003; Uhlenbrook and Hoeg, 2003). They deliver valuable information if the decisive parameters, such as

uncertainties and end-members, are properly considered. The main challenge in studies using this tool concerns the avaliability of the end-members identification and the definition of their signatures. Regarding the first point, in this study, where the identified end-members match the geological reservoirs, it must first be demonstrated that the geochemical signature of water in the different geological reservoirs differs significantly according to the geology. The approach chosen in this article to address these issues is multifaceted. It is based on a geological analysis of the groundwater collected in the basin consolidated by two

complementary approach. The first one is based on rock lixiviation, which validates that the defined end-member signatures are sounds and proves that the springs collected in the formations reflect well the formation's geochemical signature. The second





is based on a supervised classification that allows validating that the end-members are distinguishable by the geology of their reservoir.

The definition of the correct geochemical signature of the different poles is complicated by the seasonal variation of the concentration in groundwater. This increase in seasonal concentration observed in groundwater can be explained by a decrease in precipitation leading to both a decrease in the dilution process of groundwater and a possible increase in the residence time of water in the reservoir and thus an increase in concentrations. A standard method, used in most studies, focusing on flood events, recommends using extreme values to characterise the signature of each pole (Ali et al., 2010; Christophersen, 1992; Correa et al., 2019; Genereux, 1998; Iwasaki et al., 2015). However, for the groundwater reservoir with the highest mineralisation, if

only the high extreme values are considered to define the signature of the end-member, the amount of water contributed by the less mineralised water of this reservoir, i.e. with a higher dilution and shorter residence time, would be underestimated in the mixing part of the calculation. This would lead to an underestimation of the contribution of these reservoirs in terms of volumetric flow. Conversely, for the reservoirs with the lowest concentration of dissolved elements, the choice of the most diluted end-member would lead to an overestimation of their contribution to the volumetric flow. Furthermore, the natural

variability in the geochemical signature of different water samples taken from the same formation or leachate illustrates some heterogeneity in the geological formation or the weathering conditions and the need to consider a more appropriate value for defining end-members rather than a maximum or minimum.

In response to this issue, we test four different methods to define the end-members' signature and assess their relevance. As a reminder, the first two are based on the analysis of collected groundwater defined as representative of a specific reservoir, one

considering the "seasonal mean" of the groundwater geochemical signature as an end-member, and the other takes into account the geochemical signature of the groundwater samples, collected at roughly the same time as the modelled surface water. The third method considers an average signature on all groundwater samples from the geological formation over the entire watershed, the "Geological Mean" approach, and the last one is based on the results of the rock leaching experiments, the "Leaching" approach. It appears that the methods based on rock leachate analyses and the "geological mean" present structural

limitations. Regarding the "Leaching" approach, the shortcomings are linked to the limestone leachate experiments, with leachates showing significantly lower mineralisation than that observed for the groundwater, for the limestone rock formation due to a close system concerning the gas $CO_2$ during the leaching experiment, imposed by the experimental conditions. Regarding the "geological mean" method, the over-representation of the data on water collected during the pre-campaign period, between March and May 2018, i.e. in a hydrological situation of low average flow, induces an underestimation of the

mineralisation of the end-member and an increase of the standard deviation. This leads to high variability in the obtained results and their uncertainties.

The other two approaches, the "Time Window" and the "Seasonal Mean" approach, give very concordant results. However, slight discrepancies appear in the modelled parts of mixing (see Figure 11). This is especially visible in the second part of summer 2018 when the "Time Window" method differs from the "Seasonal Mean" method showing a minor decrease in the

black mica schist reservoir contribution and a minor increase in the quartz mica-schists reservoir contribution. This discrepancy may result from the impact of a heavy rainfall episode that fell at the beginning of August on the basin (about 30 mm), inducing



a visible dynamic after this event. This result suggests that the average seasonal method would not consider certain variations during the low-water period due to its excessive smoothing of the poles. Therefore, the 'Time Window' method would allow for results with greater temporal precision. Moreover, the absence of considering the seasonal variations of the end-members leads to an overestimation in low-flow of the mixing proportions of the reservoirs with a greater seasonal increase than the others. Despite the greater fluctuations for the "time window" method, it gives visibly finer results and allows a fine understanding of temporal dynamics.

Based on those observations, it can be recommend using the "Time Window" approach to identify the signature of end-members in a context of significant seasonal variability. The other approaches allow for assessing the trends but are not precise enough to compute the precise part of mixing.

## 4.2 Discussion about results

The study of the contribution at the level of the watershed's outlet or, more generally, over the whole watershed shows the importance of black mica schists during low water periods. In marge of the low water periods, it can be seen that the majority of the water flow comes from the quartz mica schists. Nevertheless, the contribution of the black mica schist reservoir remains very important considering its small surface area of only 20 % of the catchment area. The drying curves of these two reservoirs are very different, reflecting two very different behaviours with a much steeper slope and demonstrating a much lower water production capacity in low water levels for the quartz mica schists during the low-flow period. The specific flows calculated with the outcropping surfaces of each geology are less than 1 l/s/km$^2$ for the quartz mica schists and more than 2 l/s/km$^2$ for the black mica schists. All this underlines the importance of the black mica schist reservoir supporting the low water levels, which is even more marked when the flows are lower.

The analysis of the spatial distribution is in agreement with the location of the reservoirs and provides relevant results on the distribution of the productive reservoirs. We can see that the black mica schists are the biggest contributors, and the main resource area of this formation comes from the upstream part of the catchment. This result may appear contradictory due to the absence of outcrops of this formation in this part but can be justified by the presence of this formation under the limestone plateau (Arnaud, 1999). Other factors support this assertion: on the slopes where the black mica schists are inexistent, the flow rates are much lower than on the rest of the basin, and almost all the tributaries dry up during severe low water. These results allow the clear identification of the main reservoir in the low water support and could be used to guide stream water management in this catchment area to preserve the resource of this essential reservoir.

These robust results in the contribution consolidate the conclusions made by other authors who highlight the importance of groundwater in the hydrology of mountain areas (Gabrielli et al., 2012; Hale et al., 2016; Uchida et al., 2006). Nevertheless, the significant contribution of groundwater from metamorphic rocks in the basin is in contrast to traditional hydrological assumptions that consider such basement rocks in mountainous regions as having limited aquifer potential (Younger, 2007). However, there are significant differences between both schist reservoirs, with overall higher contributions from the black mica and lower contributions from the quartz mica schists. The analysis of the tributary contributions highlight an ever great variability linked to the upstream-downstream and south side north side oppositions. These show that the flow is mainly





produced during low-flow on the southern slope and, more precisely, on its upstream part. The contribution is mainly from the black mica schists in this upstream zone. One another tributary (the Valat des Oules) has a very high specific flow (1.7 l/s/km$^2$), with a contribution comming mainly from the quartz mica schists. This singularity lends credence to another hypothesis in which this difference in low-water productivity comes from a difference in weathering in the mica schists. This difference in

alteration would give the more weathered rocks a greater storage capacity and higher productivity at low water.

This higher productivity of the weathered zone has been shown in other studies (Floriancic et al., 2018; Mwakalila et al., 2002; Smith and Patton, 1981; Witty et al., 2003). It is shown that these weathered zones (e.g. saprolite or other regoliths) can serve as a larger baseflow maintenance reservoir than the underlying bedrock (Smith and Patton, 1981; Witty et al., 2003). This possible predominance of the weathered zone causes complications in interpreting the influence of bedrock type on baseflow

due to the difficulty of separating it from the contribution of the unweathered zone (Mwakalila et al., 2002). It would be relevant to test methods to differentiate these contributions such as the investigation based on the lithium isotopes ((Millot et al., 2010)). Others may be considered, indeed, a more important fracturing of a rock may cause great differences in contributions (Uchida et al., 2006), or the orientation of the schistosity plane of the layers oriented mainly towards the north (Arnaud, 1999) which can lead to more important storage of the reservoirs on the southern slope and more rapid draining of the groundwater from

the northern slope.

## 5 Conclusions

The results presented in this article are convinsing. They show that the use of tracers, as basic as major elements, revealed to be relevant to identify the contribution of the different geological reservoir to streamflow during a low-flow period in small catchment areas. The method using groundwater major element analysis of each geological reservoir to characterise the end-

members leads to sound results and validation by statistical analysis, and rock leaching analysis provides robustness to the end-members characterisation. Hence, the paper's first objective is validated: to identify and characterise the contributors to the stream flows based on simple major element analysis.

The second objective relates to the quantification of the contributions of each identified end-member. The different approaches used to characterise the geochemical signature of the end-member, i.e., "time window", "seasonal mean", "geological

mean", and "leaching", lead to comparable results. The distinction of a specific geochemical end-member associated with each geological reservoir and the measure of discharge rates allows us to quantify their contributions to the river flow. The results outline the discrepancy between the outcropping surface area of each geological reservoir and its contribution in terms of flow to the river.

It can be seen for this catchment area that the black mica schists reservoir become predominant during the e low-flow ,

although it shows only a relatively small spatial coverage. Moreover, the spatial analysis of flow contributions shows that the main contribution of this formation comes from the upstream part of the catchment where this formation hardly outcrops. Therefore, we can foresee a relatively large cover reservoir of this formation on this part of the catchment. These results





highlight the key role of this reservoir and alert the stakeholders on the need to efficiently manage and preserve these specific water resources, especially in increasing pressure and climate change.

These encouraging results were probably facilitated because the basin is relatively simple from a geological perspective and shows very little anthropic activity that could significantly impact the river's chemistry and complexify the analysis. It would appear relevant to trial this method on more complex catchments and/or those with a higher anthropic impact. The results of this study underline the predominance of a reservoir, with a small spatial extent in the support to low-water periods of the basin as a whole. They highlight the importance of a greater understanding of the functioning of watersheds at low-flows to develop

a better strategy for the management and preservation of the resource because of future climate trends.

**5.1**

*Author contributions.* Conceptualization, M.G., C.L.G.L.S., P-A.A. and P.M.; methodology, M.G., C.L.G.L.S. and P-A.A.; software, M.G.; resources, S.K., P.M. and P.V.; data curation, M.G.; writing—original draft preparation, M.G.; writing—review and editing, C.L.G.L.S. and P-A.A.; visualization, M.G.; supervision, C.L.G.L.S. and P-A.A. All authors have read and agreed to the published version of the manuscript.

*Competing interests.* The authors declare no conflict of interest.

*Acknowledgements.* The authors would like to thank all the stakeholders (administrations, owners,...) who allowed access to the sites and all the technical persons who contributed to the collection of samples during the summers of 2018 and 2019. Special thanks to the technicians and trainees of the UMR ESPACE 7300 CNRS for their support in our experimental approaches. We also thanks the communes of Moissac-

Vallée-Française and Moleizon, which gave us access to the groundwater of their communes and which performed sampling during the both campaigns





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
