# Peer review of "IDENTIFICATION OF THE CONTRIBUTING AREA TO RIVER DISCHARGE DURING LOW-FLOW PERIODS"

_Hydrology and Earth System Sciences, 2021_

## Author Response (AR1)

**Response to the reviewer 1**

The authors applied an end-members analysis to identify and quantify stream contributions during low flow period in a headwater (<100 km2) Mediterranean catchment relatively poorly affected by human activity. The originality of the work are : (1) to focus the stream flow deconvolution to the low flow period which has been rarely done but seems relevant to understand the processes that sustain low flow in a context of global changes and of a Mediterranean climate where drought are known to increase already. (2) to compare and combine different methods for identifying the end-members: hypotheses-driven identification consists in associating an end-ember to each geological unit present in the catchments. The data-driven approach supports the first identification by applying a clustering and a principal component analyses to a set of groundwater samples collected throughout the catchments and for which concentrations in major elements have been analysed.

Four methods are conducted to infer the concentrations of each members: one use the temporally closest analyses of groundwater samples, another use the seasonal mean of these groundwater concentrations, a third uses the geographical mean of groundwater concentration by geological unit and the fourth experimentally produces a leachate of each geological unit and analyses its concentrations in major elements. The Four methods are compared in terms of end-members contributions and in terms of uncertainties.

In overall, I found the study well-designed and scientifically relevant for a publication in HESS. The result highlights the dominance for sustaining low-flow of a groundwater reservoir that has a small spatial extend and a relatively moderate/low contribution to the total flow. The approach combining different methods for end members identification and quantification (including the associated uncertainty) is applicable to other sites and seems relevant to link water contributions to potential geological reservoirs.

In details, I have some questions and several suggestions to improve the clarity of the description (please see below) that should lead to relatively minor edits of the manuscript. My major critic would be that the article should include a short overview of the limits of the EMMA approach widely discussed in the literature (such as the assumed stability of the end members chemical composition and the need for objective methods to fix the number of end-members). This would not decrease the impact of the study as some of these limitations are either addressed here or less limiting because of the focus on low flows... I am not a native English speaker and this has to be checked with one but I think that the manuscript requires a little work for polishing the English.

The article was sent to a native English-speaking translator for correction and improvement of the English. A paragraph has been added in response to your comment in the introduction:

« *In order to take into account the limitations raised in the use of EMMA, includes the assumption of the conservative behaviour of the end-member tracers in the model, and fulfil the need to implement an unbiased method to define the end-members in terms of their number and the accuracy of their signatures, the approach presented will combine different tools (Barthold 2011, James 2006, Hooper 2001, Hooper 2003).*

*To take into account this limitation, it will be based on an approach will include statistical classification, a leaching approach and an independent study of the end-member signature..* »

1) **L.15, p1 Aubé, 2017, can you provide more references?**

7 references have been added ( Bard et al., 2012; Giuntoli et al., 2015; Marx et al., 2018; Ruiz-Villanueva15et al., 2014; Sauquet et al., 2015; Van Vliet et al., 2013; Vidal et al., 2016).

2) **L. 18, p1 add a space after "scarcity"**

Space added.

3) **L. 26, p2 why "However"?**

However has been removed.

4) **L. 31-32, p2 "due to the fact that many low flow studies work on small basins with a strong geological homogeneity": I think that the sentence is ambiguous. Studies investigating the origin of water have been rarely applied to low flow because hydrologists usually want to understand the processes of flow generation and therefore focus rather on the high flow periods or events. Water origin can be a relevant question even in catchments with homogeneous geology.**

This sentences has been removed.

5) **L. 79-81, p. 3, information has already be given.**

The paragraph has been removed.

6) **L. 88, p. 3: correct "Montpellier" and "Figure 1)**

The modification was made on the figure.

7) **L. 89, p3: I did not understand "This typical watershed of the Cévennes area". What is typical from what?**

I remove typical which was unnecessary.

8) **L. 91, p.4: Remove "The" before "Mediterranean"**

"The" has been removed.

9) **L. 97, p. 4: Considering that altitude ranges between 250 to 1100 m is only one rain station enough? How does rainfall vary spatially?**

Precipitation varies with altitude but this remains limited. The comparison of the average annual rainfall between Barres des Cévennes (at the top of the basin) and Saint Jean du Gard (located at 160 m on a neighbouring basin) shows slight differences of less than 100 mm. Furthermore, the purpose of the sentence was only to present the climate of the catchment area in a very broad way. I have preferred to present this in a global way rather than extrapolating these dynamics only observable in the neighbouring catchment areas. Nerveless precipitation data are presented as background information and are not further use in this paper.

10) **L. 98, p. 4: "mica schists 1"?**

« of mica schists *(Figure* 1 *).* »

**11) L. 111-112, p.4: which type of sensor is used to monitor the stream discharge? did you use a rating curve? What is the monitoring time step?**

The following sentence has been added to the text

Pressure probes are used for limnimetry measurements. Measurements are taken at a time step of 5 minutes and a rating curve is then used to calculate the flow rate from the probe measurements.

**12) L. 117, p.4 "rainfall events have a very low impact…" the authors should precise "during this low-flow period" I guess?**

During low flow period has been added.

**13) L. 130, p. 6 +L. 134, p.6: please use the standard symbol "pH" instead of "PH"**

The modification was made in the text.

**14) L. 133, p.6: why did you use a threshold of 0.25 μm for filtering? I think that usual thresholds are 0.45 and 0.20 μm used for particles and colloids respectively? Can you provide a protocol reference?**

This is a mistake, 0.45 filters were used. The change has been made in the text.

**15) L. 134, p.6: "stored until analysis" can you describe briefly in which conditions?**

The samples were stored in a cooler with ice packs.

"Tubes for the cation analysis were acidified to pH 2 with a drop of nitric acid titrated to 0.5 N and stored *in cold place* until analysis *done within 24 hours*"

**16) L. 135, p. 6 can you provide a reference for the analysis protocol (for major ions concentrations)?**

These sentences have been included:

*« The mobile phase was prepared in 1 L of deionized water (18.2 MOhms-cm at 25 °C) with 50 ml of Na2CO3 /NaHCO3 at 64mM/20mM for the anions and 25 ml of 2.6-Pyridinedicarboxylic acid at 0.02 M and 2 ml of HNO3 3N for the cations. The chromatographs obtained were calibrated according to a series of standards ranging from 0.01 to 100 mg/L for the target ions. Two control samples, one with low concentrations close to water found in metamorphic waters (EC of 50 μS/cm) and the other with high concentrations close to water from sedimentary reservoirs (EC of 600 μS/cm), water, were analysed at the beginning of each series of analysis as well in order to ensure the absence of instrumental contamination or drift. A verification step was carried out on the integration of the chromatographs obtained. »*

**17) L. 137, p.6: what are the "two representative sites"? and what are "each reservoir"?**

Modifications has been added to make the sentence more understandable. "For monitoring the low-flow period *and to respond to organisational issues*, the observation site was downsized to two representative sites for each *geological* reservoir *identified as potential end-members*"

**18) Table 1 p. 7: Could you report the ID of the sampling points in Map of Fig 1? (at least the stations that are monitored weekly) ?**

The modification was done in the Figure

**19) L. 140, p.7: "sites" plural**

The modification was done in the text

**20) L. 148, p.7 : " on several sections (4) " : are there four sections in the spatial analysis ?could you locate them in Fig. 1?**

The modification was done in the Figure

**21) L. 152, p. 8: "also collected for analysis (Figure 1). An additional campaing was…" instead of "for anlysing (Figure 1). In addition to monitoring…."**

The modification was done in the text

**22) L. 155, p. 8: "discharge measureMENTS WERE carried out by SALT DILUTION METHOD on the tributaries and BY exploring the velocity…"**

The modification was done in the text

**23) L. 160, p. 8 "analyses"**

The modification was done in the text

**24) L. 165, p.8: "End-memberrs" capital mark**

The modification was done in the text

**25) L. 167, p. 8: "principal component analysis WAS applied"**

The modification was done in the text

**26) L. 173, p. 8: Please revise English "To confirm the defined hydrogeochemical end-members"… "It aimed (past) to strengthen the previous definition of the end-members by an inverse approach"?**

The modification was done in the text

**27) L. 176-177, p. 8 : present tense is used instead of past**

The modification was done in the text

**28) L. 179, p. 8: could you provide some details about the rock samples collection method?**

Modification have been done. "For this purpose, three rock samples were collected in each of the identified geological units *in different location. The rock samples were extracted from the bedrock and all had to be larger than 10 cm sized blocks. Each sample was then stored individually until analysis*."

**29) Figure 3: In title replace "experimentation " by "experiment", in legend "Granite" with "e" , add also please the number of samples for each substrates : Limestone (3), Granite (1) …. And add the number of replicates**

The modification was done in the Figure

**30) L. 196, p. 10, remove the point after "alkalinity"**

The modification was done in the text

**31) L. 200, p.10: "one of the objectives WAS"**

The modification was done in the text

**32) L. 202, p. 10: "the number of required tracers WAS… "**

The modification was done in the text

**33) L. 206, p. 10 "End-member…" capital mark**

The modification was done in the text

**34) L. 208, p. 10 "our approach USED…"**

The modification was done in the text

**35) L. 218, p. 10: I did not understand here? How much parameters did you use?**

The parameters have been left as defined by delsman in his guide to using the model. Only the number of iterations has been changed.

**36) L. 222, p. 10: "The objective WAS…"**

The modification was done in the text

**37) L. 241, p. 11: I missed in this section the quick description of the software and packages used here (for PCA, clustering, EMMA and GLUE analyses)**

Two sentences has been added:

- "*The k-means analysis was done on R with stats packages*".

- « *This PCA was done on R using FactoMineR package*".

A description of the mixing model is already given.

"The End-Member Mixing Analysis (EMMA) was chosen to assess the contribution of the different geochemical end-members identified. Our approach used EMMA coupled with the Generalised Likelihood Uncertainty Estimate (GLUE), called G-EMMA and developed by (Delsman., et al, 2013}. This GLUE method, developed by {Beven., et al, 1992}, manages uncertainties by accepting variation in sets of input parameters. A full range of plausible results can be explored with model executions within a user-defined range by varying the input parameters. The G-EMMA method considers both the uncertainties in the conceptualisation of the model (validity of the choice of the end-members) and the measurement uncertainties related to the analytical errors. The variability accorded to the tracers chosen for the surface water is defined by the uncertainty associated with the devices used in the measurement (5 \%).

**38) L. 243 to 251, p. 11: is this paragraph new? useful? necessary?**

This paragraph was removed.

**39) L. 254, p. 11 + L. 270, p. 12: "Piper diagram" with a capital mark**

The modification was done in the text.

**40) L. 254, p. 11: "are identified" visually?**

Visually has been added.

**41) L. 256, p. 11 : "with the composition of 3 groundwater"?? the three? Which?**

It was a mistake 3 has been removed

**42) L. 276-277, p. 11 : Is there a missing end-member then?**

The granitic section has very little extension, so its contribution can only be a minor part of the whole run-off generation and it's consistent to not find its signature in the groundwater. Moreover the rare springs in the granitic section have a signature corresponding to the mixture of groundwater from the limestones and the black micaschists. Hence for all this reasons the choice was made to disregard this end member.

The followed sentence has been added to clarify this point:

"*For these different reasons and the very small extension of the granitic part on the watershed, this reservoir was not considered as an end-member.*"

**43) L. 279-280, p.11: Figure 5 does not show seasonal variation, does it?**

This seasonal variation can be observed in figure 5, which shows a progressive increase in the concentration of ions (Ca, Mg, Na, SO4). The points with the lowest concentrations correspond to the beginning of summer and those with the highest concentrations to the low water period.

Arrows has been added to the figure 5 with a sentence in the caption: « *The arrows mark the seasonal variation of the different geological reservoirs.* "

**44) L. 285, p. 11 : "close to the observed signatures from groundwater samples"**

The modification was done in the text

**45) Figure 6: please find a transparency or something to make the GW sample less visible compared to the leachate.**

The modification was done in the Figure. Symbols of groundwater were downsized and had made more transparent.

**46) L. 303, p. 14: not shown?**

The choice of not showing the concentrations of K and Ni ions was made to facilitate the reading of the graph and limit the number of panels.

**47) Figure 7: caption : "Silhouette Values to define the optimal numbers of clusters"**

The modification was done in the text.

**48) L. 313, p. 15 : "inertia curves..on the thre classes" ?? I do not understand**

The sentences was modified: "*Inertia curves define an optimal value of 3 classes, and* gives equivalent results to previous analyses on groundwater samples to characterise the end-members ".

**49) Table 2 caption "clusters"**

The modification was done in the text

50) **L. 336, p. 16 "due to their low frequency of detecting…" replace by " because often below the detection limits" ?**

The modification was done in the text

51) **L. 337-339, p. 16 formulation looks strange, isn't it?**

The sentences was modified: "Due to *the* low concentration of *total dissolved solids* in *all of measured dissolved ionic elements in the groundwater from quartz mica schist reservoir*, no tracers *were* specifically identified for this reservoir.  This reservoir acts as a dilution *end member* for all tracers"

52) **L. 340, p. 16 "to add tracer to the tracers chosen by the end-members" : what do you want to say here?**

The sentences was modified: " To improve the efficiency of the model and to conform *and follow the methodology developed in Barthold  et al (2011)*, the choice was made *to add one additional tracer*"

53) **Figure 9: more contrasted colors would be easier to read**

The Figure have been modified. The colours have remained unchanged but the size of the item has been increased

54) **Table 3: "pH" instead of "PH", what is illustrated using bold characters in the column/line titles? In the table itself?**

The modification was done in the Table and bold character has been removed.

55) **L. 349, p. 18: the collection of rainfall water should be presented in the material and method section, at this stage we do not know that some rainwater has been sampled**

A paragraph has been added in the end of section sampling and analysis :

*"Rainwater samples were collected using the same methodology as for groundwater. The water was collected from a rain gauge located in a neighbouring catchment area less than 10 km south of the catchment."*

56) **L. 354, p. 18 precise in the subsection title "of time-window method "**

Modification was done in the title.

57) **L. 361, p. 18 what does mean "minus 10%", is it inferior to 10%?**

Minus was replaced by "under"

58) **L. 365, p. 18 "coherent with the increasing tourism activity" , I don't think so : usually WWTP rejects are constant throughout the year (unless the tourism increases drastically the population in summer?), but this constant reject is less diluted in summer low-flow period because natural stream flow is lower…**

I agree with you on the predominance of the decrease of the water flow in the increase of the contributions of the wwtp however the catchment area is very little populated and the Cevennes are a rather tourist area with a strong increase of the population in summer. I do not have the numbers

but I think that the population increases by at least half during the summer. I have changed the sentence

"A more important contribution of WWTP can be observed from mid-July to the end of August, coherent with the *decrease of natural stream and the increase of WWTP effluent due to the increase of population during summer* but remaining nevertheless below 4 \%."

**59) Figure 10: please add the associated uncertainties, is 2018 in upper panel and 2019 in the bottom panel? If so it should be written in the caption.**

The uncertainties has been added in caption:

*"The uncertainty associated with these proportions is less than 15% for WWTP and limestone waters and less than 35% for quartz and black micaschist waters."*

**60) Figure 11: "black represents the contribution of WWTP"? It appears "yellow" in my pdf**

"Black" has been replaced by "*yellow*" in the legend

**61) L. 414, p. 21: where can we see the conductivity of these higher/lower contributions?**

The average conductivity of the different reservoirs was given in the presentation of the end members (400 for the limestone waters, 100 for the black micaschists and less than 60 for the quartz micaschist waters). Modification has been done in the sentence :

**"**Regarding the "seasonal average" output, the results show a lower contribution of the waters with the highest low-flow electrical conductivity *(Limestone)* and a higher contribution of the waters with the lowest *electrical conductivity (Quartz mica schist)."*

**62) L. 417-421, p. 21 : These sentences should be in the Mat and Method section**

A paragraph present already this campaign in mat and method section. It's just a reminder, should I remove it.

"An additional campaign was carried out in 2019 to analyse the spatial contribution of tributaries to the main watercourse throughout its route. Gauging and sampling were performed on five sites distributed along the main river, and six tributaries were targeted (3 per side) using the same sampling and laboratory analysis method presented above. The discharge measurements were carried out by salt dilution method on the tributaries and by exploring the velocity field using a current meter for the main watercourse. The operation aimed to analyse the contribution of the reservoirs with a spatial approach. However, only one tributary on the northern slope could be analysed as the two others were dry."

**63) 426, p. 21 : "less than 0.1 l/s/km2"**

The modification was done in the text

**64) Figure 12: using same colors for geological map and flow contributions is ambiguous, 7 is not readable, 3 is not numbered**

The number has been changed but I think the choice of colours is logical and understandable even if it may make the diagrams less readable on an identical background

**65) L. 450, p. 23 : what concentration does increase?**

Modification was done in the text:

« This increase of *different ion concentration (Ca$^{2+}$, Mg$^{2+}$, SO$^2_4$ , HCO $^{2-}_3$) during summer* observed in groundwater can be explained by a decrease in … »

**66) L. 456-459, p. 23: Nevertheless, as here you try to identify low flow contributions it remains relevant as low flows are rather associated to the period where contributions are poorly diluted and residence times highest?**

At the end of the low water period this is correct but the article refers to the whole summer low water period. During the beginning of this one the concentrations in ions are much lower thus more diluted and evolves gradually what justifies the assumption of a signature taken on a value close temporally and not the extreme seasonal value.

**67) L. 475, p. 23: How do you explain the increase of standard deviation here?**

This increase in uncertainty is for me linked to a lower quality of the end members definition. Taking all the measurements made, including the spring one. The groundwater signal is more diluted and therefore gives a wider range of possible concentration for the poles which gives a wider range of results and therefore a stronger standard deviation.

"*(caused by a wider range of results)*" has been added after standard deviation

**68) L. 497, p. 23: Precise when this specific flow calculations have been conducted (to which period are they associated)**

The modification was done in the text

**69) L. 517, p. 25 : Does "The Valat des Oules" correspond to a number provided in Figure 11?**

Yes, the number has been added in the text.

**70) L. 523-525, p. 25: then the Quartz and black micaschists reservoirs are supposed to be fissured layers of micaschists only without contribution of a weathered layer?**

This is a possibility. The contribution of the tributaries is different for quartz mica basins, which could support this hypothesis. But it can also be due to differences in the schistosity plan (allowing higher capacity) or other explanatory elements. The data do not allow us to decide on these points.

**71) L. 535, p.25 "analysis provided"**

The modification was done in the text

**72) L. 541, p. 25: "allowed us"**

The modification was done in the text

**73) L. 552, p. 25 : did you mean a higher anthropic impact on the water quantity or on the water quality?**

On the water quality.

**Response to the reviewer 2**

This study estimates the contribution of several groundwater reservoirs, differentiated by geological characteristics, to the low-flow period of a Mediterrenean catchment. The authors used various methods to decide on the end-members in an EMMA, and considered correlations between tracers and their variability. I appreciate this thourough analysis and relatively objective decision on end-members, and the work the authors put in here. Additionally, the uncertainty analysis was well performed by using different end-member definitions.

However, I must admit I stopped reading the manuscript shortly before the Discussion sections. The manuscript is overflowing with English mistakes, mistakes in units, inconsistencies, strange sentences, etc. It is very difficult to follow and URGENTLY needs a native speaker to polish it up. It borders on unreadable, and it takes too much concentration while reading. Sometimes I was not even sure if I understood a sentence correctly and I had to read it five times; this must not be the case in a scientific publication.

Besides this major point, the results section sometimes includes discussion sections...either the authors separate these more clearly, or they combine results and discussions. I have no preference here.

Following is a list of incosistencies and questions I had up to approximately the Discussion section. I hope that in a future version of this manuscript, the readability is strongly increased.

The article was sent to a native English-speaking translator for correction and improvement of the English.

**Title: no caps lock, and change "period" to "periods"**

Periods has been changed.

**L2: change "trial" to "evaluate"**

The modification was made in the text.

**L5: delete "the" before "trajectory"**

The modification was made in the text.

**L16: delete the comma before "and quality" and put it behind "quality"**

The modification was made in the text.

**L18-19: the sentence closes in on itself: "investigating processes that sustain streamflow are required to understand processes that maintain low-flows" (with sustain and maintain, and streamflow and low-flow having basically the same meaning). The sentence says "investigating A is necessary to understand A", which is logical and does not need to be mentioned.**

« Investigations on the process that sustains streamflow have been identified as a requirement to understanding *the dynamics of the hydrological system* »

**L27: "maintaining base flow" … also only in mountain areas? Also, in this line "base flow" changes to "baseflow". Decide on one spelling.**

The change was made in the text and the baseflow was kept.

**L30: delete first "geological"**

Geological has been deleted.

**L32: the aim of which study? (of course yours, but needs to be mentioned, especially after discussing a lot of other studies before)**

The modification was made in the text.

"but this has rarely been applied to low-flow. The aim of *this* study was, therefore, to identify and then quantify the contributions of the different geological reservoirs during low water conditions in a watershed showing a variety of geological facies."

**L42-44: unclear sentence. What is the main interest of EMMA? What are model output probabilities?**

The modification was made in the sentence:

"The main interest of the EMMA analysis *resides* in the ability *of this model* to consider the *whole* dispersion of *the* tracers and thus consider*ing* all possible mixing configuration associated with *their* output probabilities *calculated in the runs of the model.*"

**L45-48: what is made possible by differentiating water by season, etc.? The combination of hydrometric data and hydrogeochemical data, of the previous sentence? Or does the EMMA method make differentiating water by season possible?**

The EMMA method make differentiating water by season possible and modification was done in the sentence to clarify it.

"*The use of this model with geochemical and hydrological data permits the decomposition of the discharge in several ways. It is thus possible to quantify the proportion of water coming from different seasonal recharges or to quantify the proportion coming from different units of the discharge.*"

**L50: which assumptions, of which tools? Unclear.**

The modification was made in the sentence:

Uncertainties in the contribution estimation obtained with these models can only be minimised if the assumptions made for these *tools (use of non-reactive tracer and marked difference in the end member)* are followed

**L58: which water table? Groundwater? Why is it limited? Is it artesian groundwater?**

It's the alluvial water table. The modification was made in the sentence:

"By focusing only on low water *period* in a watershed where the *alluvial* water table is limited,"

**L60: "this paper search"?? applicability of which methods?**

The EMMA method which the last paragraph refers to. Changes have been made to make the sentence clearer.

"This paper *shows* the applicability of *the EMMA method* for identifying the origin of surface water during low-flow to understand flows dynamics in catchments during scarcity."

**L62: "productivity" is a strange word for runoff-generation contribution here.**

Productivity has been replaced by *runoff-generation contribution*.

**L72: why does focusing on low-flows allow the increase of the sampling rate? Also when sampling the whole hydrograph, the sampling rate can be increased, e.g., with automatic samplers.**

This is indeed possible. However, the analysis made only on the summer low water allows to limit the number of analysis which facilitates a more important frequency. In the case of full hydrological cycle studies the majority of studies adopt a monthly frequency with a focus on flood events. Allow was replaced by *facilitates*.

**L73-77: delete. This is just a summary of the basic layout of every research paper, and therefor unnecessary.**

This paragraph was deleted.

**L88: Figure 1 1? Figure 1 l (L)?**

The modification was made in the text.

**L91: 1,110 mm. Adapt everywhere and avoid the space between digits.**

The modification was made in the text.

**L93: 50 mm in which months exactly? Jul-Sep? And autumn from Oct to Dec?**

Though total rainfall is high, summer rainfall is very low, less than 50 mm *(July to September),* and almost half of the total annual rainfall falls in autumn *(October to December)* during high-intensity rainfall events.

**L94: modulus = ? also mean monthly annual discharge without capital letters, and the term is very confusing. Is this the monthly or annual discharge? Is this the mean minimum discharge of all months for all years?**

The sentence has been changed:

"The Gardon de Sainte-Croix river has a *mean annual discharge* of 960 l/s, and its *m*ean *m*onthly *a*nnual *m*inimum discharge is equal to 0.135 l/s at the hydrometric station located at approximately one-third of the basin length …"

**L98: "mica schists 1"? what is the meaning of the "1"?**

The sentence has been corrected:

"mica schists *(Figure* 1*)*"

**L105: "and suitable to trial our approach carry out our research" English needs correction**

The sentence has been corrected:

"Hence the basin can be considered as little affected by human activity and *suitable to trial our approach.*

**L111: I would either choose l/s or m³/s to make it easily comparable with line 94.**

The flow rate has been harmonized in l/s.

**L117-118: The first sentence says that rainfall has a low impact on runoff, while the next sentence starts with the information that runoff variations are due to rainfall events.**

The whole paragraph has been changed.

*"It can be shown that the importance of the volumetric discharge rate at the beginning of the monitoring period is linked to the amount of rainfall during winter and spring.  But during the summer period the rainy events have a low impact on the stream volumetric discharge rate which shows small and brief peaks following these events. It can be seen that the flow returns to a level lower than that of the flow measured before the event in 1 to 3 days. This implies that the recharge brought by these rains to the subsurface reservoirs is negligible and makes it possible to neglect its impact on thoses reservoirs in our future modelling."*

**L128: call hydrogen potential by the more commonly known pH. Like in L130, but not PH but pH.**

pH has been corrected in the whole document.

**L132: collected in tubes? Not in sample bottles? If tubes, how were the closed?**

They were collected in plastic closed tubes to be analysed the same evening and to limit the loss of nitrates and simplify handling in the field. A spare sample bottle of sample was also collected to be preserved and analysed in case of suspected errors in the analysis.

"Samples for the analysis of major ions were collected in *closed* polyethene tubes *suitable for analyses on the IC* (one for the cation and one for the anion)."

**L137: which reservoirs? It is unclear how the sub-catchments were chosen, since what is meant with reservoir is not clear and Table 1 is not helpful in this regard.**

The mention "*identified as potential end members*" has been added to clarify the sentence.

"*To facilitate the* monitoring *during* the low-flow period, the observations *were* downsized to two representative sites for each *geological* reservoir *identified as potential end members*"

**L147: what is the difference of river and surface water? Tributaries? Lakes? And what was the sampling frequency for surface waters?**

The sentence has been changed:

"The 2018 campaign focus on the characterisation of the groundwater contribution during the drying up period of the river with a *high* frequency, *weekly sampling* for surface water and bi-monthly for groundwater."

The term "river" is used for the main river (Gardon de Sainte Croix) and the term "tributaries" is used for the tributaries of this river. There are no lakes mentioned in the publication.

**L148: does the 4 in brackets mean that four sections were sampled? Why not directly write this then?**

Yes and modifications has been done to clarify it:

"The 2019 monitoring period was complemented to include a spatial analysis where the stream was sampled on *four* sections *(shown as: I, II, III and IV on Figure 1)* ..."

**L149: what is a "large panel of groundwater"? Also, why does the sampling campaign in 2019 last until December when in line 145 it stops in October? Is this a different sampling campaign and "completed" in Line 148 should be "complemented"?**

The mention "at least" has been added to clarify the text. And larger panel

"Two monitoring campaigns were carried out during the summer of 2018 and 2019. Both spanned *at least* from June to October;"

"… and campaigns including a larger panel of groundwater *sampling site (8 spring or boreholes sites)* were carried out every month

And larger panel has been further detailed and campaigns, with a larger panel  of groundwater *sampling site (8 spring or boreholes sites),* were carried"

Completed was also changed by complemented

**L151: the wastewater treatment plant was not mentioned in the study description where "minimal anthropogenic influence" was discussed, only tourism and a cheese factory. It must be mentioned there, especially if samples were taken from it…for which I assume it can not simply be ignored.**

Anthropic activities that can impact the stream water quality include tourism, with only two campsites, *one waste water treatment plant* and a cheese factory, all located in the downstream section of the basin.

**L154: the description of the additional sampling campaign does not make sense. Five sites were monitored with six tributaries, with 3 per site. 3 x 5 = 15, and not 6. Or is it "side", as in left and right side of the river? That would explain the 6, but why then five locations?**

The sentence has been changed to be clearer:

"Gauging and sampling were performed on five sites distributed along the main river, and *also on* six tributaries (3 *on each river* side) using the same sampling and laboratory analysis method presented above."

**L156: description of the salt tracer method applied to the tributaries needs to be rephrased, as it is almost incomprehensible. At least I assume the salt tracer method was used, since dilution gauging was mentioned, but then a current meter is used which is a sensor that measures the flow of water and not electrical conductivity…which was used in the main river and not the tributaries where actually dilution gauging was applied (?)…this section is like many before very, very confusing.**

"The discharge measur*ements were* carried out by *salt* dilution *method* on the tributaries and *by* exploring the velocity field using a current meter for the main watercourse."

**L182: use "18.2 MΩ" instead of "18.2 M". "Rock water" should just be "rock", since it's the mixture of rock powder with the ultrapure type 1 water?**

Modifications has been done.

"Rock powder was mixed with ultrapure water (18.2 M$\Omega$) in a 50 mL bottle, in ratio of 1/10 (3g rock water to 30g water)"

**L187: delete "presented in 2.2"**

"Presented in 2.2" has been deleted.

**L208: the name of GLUE is wrong, it's likelihood and not probability. The explanation of GLUE in the next two sentences is lacking.**

Modifications has been done.

**L214: the uncertainty is associated with 5% of the devices used to analyze the data? The sentence seems to say this. And how is the uncertainty due to the chosen tracer in any way related to the measurement precision? This is measurement uncertainty and not the model conceptualization uncertainty.**

There is a difference between analytics errors and natural variability, the sentence just says that the variability allowed for tracers for surface water is set at 5% which is the uncertainty associated with the measurements. For surface water, only 1 point was measured in each campaign and does not show natural variability in a single time point. The only uncertainties that can be attributed to surface water are the uncertainty of the analytical means

"The variability accorded to the tracers chosen for the surface water is defined by the uncertainty associated with the devices used in the measurement (5%)"

**L214-215: "A temporal variation treats uncertainties associated with the choice of geochemical poles." I have no idea what this means. Temporal variation of what? Treats uncertainties?**

Modifications has been done.

"A temporal variation *of ionic concentration is included in the* uncertainties associated with the choice of geochemical poles".

**L215: linking measurement uncertainty to the variation in the tracer signal makes absolutely no sense. Of course the tracer signal will vary in time, otherwise it's not a good tracer. The measurement uncertainty must be linked to the uncertainty of the measurement device, or sampling uncertainty due to spatial heterogeneity.**

The sentences has been corrected

« This measurement uncertainties are defined by the variation *in the measurements of the control samples.* »

The explanation of the « *control sample* » has been added in the methodology :

« *The chromatographs obtained were calibrated according to a series of standard ranging from 0.01 to 100 mg/L for the target ions. The mobile phase was prepared in 1 L of deionized water (18.2 MOhms-cm at 25 °C) with 50 ml of $Na_2CO_3$ /$NaHCO_3$ at 64mM/20mM for the anions and 25 ml of 2.6-Pyridinedicarboxylic acide at 0.02 M and 2 ml of $HNO_3$ 3N for the cations. Two control samples, one with low concentrations close to water found in metamorphic waters (EC of 50 µS/cm) and the other with high concentrations close to water from sedimentary reservoirs (EC of 600 µS/cm), water, were analyzed at the beginning of each serie of analysis as well in order to ensure the absence of*

*instrumental contamination or drift. A verification step was carried out on the integration of the chromatographs obtained.* »

This section presents only the variability associated with surface water tracers. The variability of groundwater is presented in the following section, which presents the different approaches used to try to take it into account (temporal monirtoring, seasonnal mean, geological mean) which integrate the spatial heterogeneity . It is therefore not relevant to address these issues of heterogeneity or spatial variability here

**L224: the digits in brackets behind each method that follows are unnecessary**

The digits are been deleted.

**L229: why does the seasonal mean (2), which considers seasonal mean values, then use annual mean values?**

The calculation has been done on the average of the groundwater sites previously defined as representative of the formation and not on the annual average. The average is therefore seasonal because it takes into account only the values during the summer low water period and not an annual average.

"*Annual*" has been removed in the sentence:

"The second method, so-called hereafter "Seasonal Mean ", consider the mean seasonal value of the groundwaters selected as representative of the reservoir."

**L254: mark the discussed end-members also clearly in Figure 4.**

Ellipse has been added to Figure 4.

**L256: "3 groundwater"???**

"This end-member is composed exclusively of water from sedimentary rock reservoirs, mainly limestones and dolomites, hence consistent with the composition of groundwater *issue from limestone aquifers* found in the literature."

**L258: correct unit is µS/cm, not µs/cm which looks like seconds.**

The modification was done.

**L260: correct to assume the authors mean MEGAequivalent per liter with MEQ/l (10^6), instead of milliequivalents per liter mEq/l (10^(-3))?**

The modification was done.

**L264: the sulphate contents remain low for other elements?**

Sentence has been changed:

"Indeed, sulphate contents vary from 0.3 to 1 mEq/l and remain relatively low for all other *end-members*."

**L265: specify the schist alterations at least in the Discussion if not here directly**

There is no specific information on schist alteration on this watershed that can be added her. But the discussion includes a section on schist alteration."

**L277: it was not mentioned in the site description that black mica schist is directly under the limestone plateau. In which depth does the limestone end and black mica schist start?**

Only one source of information (a drilling) carried out on the limestone plateau was found and it gives a thickness of 20 meters for the limestones laid on the black micaschists.

**L279: it is mentioned that Figure 5 shows a seasonal evolution of ion concentrations, but Figure 5 has no temporal information, at least not clearly discernible.**

The seasonal variation is discerned by the drift seen on the waters of the different reservoirs. A sentence has been added to better explain the observed variation.

"The definition of the correct geochemical signature of the different poles is complicated by the seasonal variation of the concentration in groundwater. *This increase of different ion concentration $Ca^{2+}$, $Mg^{2+}$,$SO_4^{2-}$ , $HCO_3^{2-}$ during summer* observed in groundwater … "

It is true that this variation is not very discernible on graphs that do not treat the information in a temporal manner, but the document already contains a lot of figures and the addition of a new figure does not seem judicious to me.

**L287: "lixiviation" –> "leaching"**

Modifications has been done.

**L322: There are more than one outlier visible in Figure 8, I count at least 3, e.g. borehole cluster 1 in black mica schist, two cluster three elements in black mica schist.**

Only two outlier are visible. The first corresponds to the point presents in the granite formation and previously identified as a mixture of limestone and black mica schist. The second correspond to the black mica schist spring identified by the classification as being from the quartz micaschist pole. Conversely, the K-means method attributes this point to the sedimentary rock clusters' in coherence with the mixing hypothesis of groundwater issue from limestone and black mica schists.

The drilling in the black micaschists is badly placed (his location has been corrected in the figure), the point is located on the fountain but the drilling is located in the limestone causse.

**L346-347: identical measurements of dissolved oxygen in springs and surface water to those of streams and springs? Two times surface water/stream and springs are compared, of course they are the same.**

The sentence has been modified by delating some term

"The measurement of dissolved oxygen in the springs confirms this by revealing identical oxygen concentrations to those found in the streams"

**L361: contribution of -10%?**

Minus has been changed *in under 10%.*

**Figure 1: in the caption "a water mine is a horizontal well dug a slope"??? what does this mean? Why is the station PR marked in the figure and not also the main outlet?**

It is the definition of this term quite specific to the Cevennes. The main output has been noted C4 and PR corresponds to the measuring station presented in the text and located on this figure

**Figure 2 does not seem to transport important information that is not also described in the text and could be deleted.**

The figure is used in the document and is used as a reference in the results section to show the differences in flow rates between years

**Figure 4: the legend is too small.**

The legend was made bigger.

**Figure 5: µS/cm is written as uS/cm. Are all axis logarithmic? Must be mentioned somewhere.**

The modifications has been made.

**Figure 9: change colors from black mica schist and limestone, they are very difficult to differentiate**

The size of the symbols have been increased to make the difference between water from black micaschists and limestones more discernible

**Table 1: more confusing than helpful. Is it important if collection was outsourced? Outsourced to whom? Can be deleted in my opinion. What do the numbers in the rows "Sampling in 2018" and "Sampling in 2019" mean? What does bold mean? The text is not helpful due to the lack in English.**

Modification has been done Sampling in 2018– Number of sampling in 2018

For bold row it was explain in the caption: "The bold row in the table correspond at main groundwater sites, sample weekly."

**Table 3: the caption mentions red values, there are no red values**

Red has been changed in bold.

---

## Referee Report (RR1)

The authors have addressed or answered all of my comments therefore I think that the manuscript is now ready for publication in HESS